# YTHDF1 links hypoxia adaptation and non-small cell lung cancer progression

Yulin Shi[1,2,11], Songqing Fan[3,11], Mengge Wu[4,11], Zhixiang Zuo [5], Xingyang Li[5], Liping Jiang[1], Qiushuo Shen[1,2], Peifang Xu[1], Lin Zeng[1,2], Yongchun Zhou[4], Yunchao Huang[4], Zuozhang Yang[4], Jumin Zhou[1], Jing Gao[6], Hu Zhou [6], Shuhua Xu [7,8], Hongbin Ji [9], Peng Shi[8,10], Dong-Dong Wu [8,10]*, Cuiping Yang[1]* & Yongbin Chen[1,8]*

Hypoxia occurs naturally at high-altitudes and pathologically in hypoxic solid tumors. Here, we report that genes involved in various human cancers evolved rapidly in Tibetans and six Tibetan domestic mammals compared to reciprocal lowlanders. Furthermore, m6A modified mRNA binding protein YTHDF1, one of evolutionary positively selected genes for high-altitude adaptation is amplified in various cancers, including non-small cell lung cancer (NSCLC). We show that YTHDF1 deficiency inhibits NSCLC cell proliferation and xenograft tumor formation through regulating the translational efficiency of CDK2, CDK4, and cyclin D1, and that YTHDF1 depletion restrains de novo lung adenocarcinomas (ADC) progression. However, we observe that YTHDF1 high expression correlates with better clinical outcome, with its depletion rendering cancerous cells resistant to cisplatin (DDP) treatment. Mechanistic studies identified the Keap1-Nrf2-AKR1C1 axis as the downstream mediator of YTHDF1. Together, these findings highlight the critical role of YTHDF1 in both hypoxia adaptation and pathogenesis of NSCLC.

[1] Key Laboratory of Animal Models and Human Disease Mechanisms of Chinese Academy of Sciences & Yunnan Province, Kunming Institute of Zoology, Kunming, Yunnan 650223, China. [2] Kunming College of Life Science, University of Chinese Academy of Sciences, Kunming 650204, China. [3] Department of Pathology, the Second Xiangya Hospital, Central South University, Changsha, Hunan 410000, China. [4] Kunming Medical University, Kunming 650223, China. [5] Sun Yat-sen University Cancer Center, State Key Laboratory of Oncology in South China, Collaborative Innovation Center for Cancer Medicine, Sun Yat-sen University, Guangzhou 510060, China. [6] Department of Analytical Chemistry and CAS Key Laboratory of Receptor Research, Shanghai Institute of Materia Medica, Chinese Academy of Sciences, 555 Zuchongzhi Road, Shanghai 201203, China. [7] Key Laboratory of Computational Biology, CAS-MPG Partner Institute for Computational Biology, Shanghai Institutes for Biological Sciences, Chinese Academy of Sciences, Shanghai 200031, China. [8] Center for Excellence in Animal Evolution and Genetics, Chinese Academy of Sciences, Kunming, Yunnan 650223, China. [9] Institute of Biochemistry and Cell Biology, Shanghai Institutes for Biological Sciences, Chinese Academy of Sciences, Shanghai 200031, China. [10] State Key Laboratory of Genetic Resources and Evolution, Kunming Institute of Zoology, Chinese Academy of Sciences, Kunming 650223, China. [11] These authors contributed equally: Yulin Shi, Songqing Fan, Mengge Wu. *email: wudongdong@mail.kiz.ac.cn; cuipingyang@mail.kiz.ac.cn; ybchen@mail.kiz.ac.cn

Hypoxia, occurring under normal and pathological conditions, imposes stress to cells and organisms. A lack of oxygen has been linked to various human disease conditions including diabetes and cancer, and constitutes challenges for mammals including humans living at high altitude[1–3]. In the past decade, many hypoxia adaptation selected SNPs have been identified from studies on genetic variants that affect the homeostatic and pathological responses to hypoxia in humans, wild and domestic animals at high altitude[4,5]. For example, positive selection has been found on prolyl hydroxylase 2 (PHD2) coding gene *EGLN1*, where adaptive amino acid changing mutations increased HIF-2α degradation to reduce the hemoglobin (Hb) concentration in Tibetans, protecting them from polycythemia, a condition considered as a blunted physiological response at high altitude[6]. Consistent with this observation, recent studies showed that the Tibetan specific HIF-2α adaptive mutation down regulates its own expression[7,8]. Therefore, hypoxia adaptation selected genes more likely play anti-hypoxia or anti-HIF1/2 dependent roles to make animals or humans behave normally under hypoxic environments. In contrast, hypoxia can activate many adaptive cellular responses contributing to hypoxic solid tumor progression, which is associated with poor clinical outcome[9]. Under hypoxic conditions, PHDs cannot hydroxylate HIF-1α and HIF-2α, which leads to HIF-1/2 protein stabilization, nuclear translocation and transcriptional activation, promoting tumor progression through a metabolic shift toward glycolysis, induction of angiogenesis, migration, and more[10]. Thus, molecular events involved in both hypoxia adaptation and hypoxic solid tumors may not necessarily behave identically.

Biomarkers for hypoxic solid tumors are traditionally identified by comparing cancerous with paracancerous tissues using comprehensive integrative analyses. Burrowing rodents like naked and blind mole rats living under extreme hypoxic conditions evolve strong hypoxic tolerance and cancer resistance, and could reveal molecular events important for cancer progression[11,12], suggests that we can use evolutionary theory to identify genes involved in both hypoxia adaptation and hypoxic solid tumors.

In this study, the large-scale population genome and transcriptome data from Tibetan domestic mammals including dog, horse, pig, cattle, sheep, and goat are analyzed, and are compared with their corresponding species from low elevations. We find that many genes involved in cancers evolved under positive selection in Tibetan domesticated mammals. We identify YTHDF1, an m[6]A modified mRNA binding protein, as a positively selected candidate gene for hypoxia adaptation. Discovered in the 1970s, m[6]A is the most prevalent type of modification for internal mRNA/lncRNA observed in a wide range of organisms ranging from viruses to yeast, and to mammals[13–16]. Recent findings have shown that deregulation of m[6]A modification leads to brain developmental abnormalities and other diseases, including cancers[17–20]. YTHDF1 expression is decreased in highland mammals compared to lowlanders, furthermore it is amplified in various types of cancers including NSCLC. We show that YTHDF1 inhibition suppresses NSCLC cell proliferation, colony formation, xenograft tumor formation, and de novo lung ADC progression. We find that YTHDF1 low expression correlates with a worse clinical outcome by rendering cancerous cells resistant to cisplatin treatment through upregulating an antioxidant system which is Keap1-Nrf2-AKR1C1 dependent, and demonstrate that the resistance of hypoxia-induced cellular apoptosis in YTHDF1 knockdown BEAS-2B cells utilizes the same axis. In summary, this study provides insights into not only adaptive evolution, but also the search for therapeutic targets for cancers.

## Results

**Rapid evolution of genes involved in cancers.** As hypoxia is a driving force of tumor progression and metastasis by influencing the expression of many tumor-associated genes (TAGs), we investigated the evolutionary pattern of genes involved in various cancers in human and six Tibetan domestic mammals (dog, horse, pig, cattle, sheep and goat) at high altitude. This was done using two groups of genes associated with cancers: one group contained all the gene mutational information causally implicated in cancers obtained from the Cancer Gene Census (CGC) database[21]; and the other group of TAGs from the PubMed database (http://www.binfo.ncku.edu.tw/TAG/) (Supplementary Fig. 1a). We first utilized genome-wide single-nucleotide polymorphisms (SNPs), genotyped by the Affymetrix Genome-Wide Human SNP 6.0 Array, from 31 unrelated Tibetans[22], and compared them with genomic data for Han people from HapMap (phase II, http://hapmap.ncbi.nlm.nih.gov/). As evaluated by $F_{ST}$[23], SNPs in CGC genes harbored significantly higher levels of population differentiation between Tibetans and Han Chinese (mean $F_{ST}$ value is 0.0596), than SNPs in other genes (mean $F_{ST}$ value is 0.0585) ($P = 3.04 \times 10^{-6}$, by Mann–Whitney U test), suggesting potentially faster evolution of CGC genes in the Tibetan population. Meanwhile, SNPs in the TAG genes also displayed a higher level of population differentiation than that in other genes, although it did not reach statistical significance ($P = 0.44$, by Mann–Whitney U test). Since the genotyping SNP array has ascertained bias, we further used whole genome sequences of Han Chinese and Tibetans from a previous study[24,25], and calculated iHS (Integrated Haplotype Score), XP-EHH (cross population extended haplotype homozygosity) and $F_{ST}$ values for each SNP to evaluate the evolution of Tibetans[23,26]. We found strong evidence of positive selection on genes associated with cancers (Fig. 1a).

Next, we interrogated the evolutionary patterns of human cancer related genes in six Tibetan domestic mammals including dog, horse, pig, cattle, sheep, and goat. The differentiation of each SNP was evaluated using $F_{ST}$, XP-EHH and ΔDAF (the difference of the derived allele frequencies) between the populations from high and low altitudes. Both CGC and TAG genes exhibited significant increased evolution rate among the Tibetan domestic mammals, which is inline with the above result in Tibetans (Fig. 1b). Because the phenotypic evolution is tightly coupled to changes of gene expression, we reasoned that TAGs might also change their mRNA expression in domestic mammals in the highland, in addition to DNA sequence alterations. To verify this, we used transcriptomes of lung tissues from four Tibetan pigs and four lowland Min pigs by RNA-sequencing. We found that differentially expressed genes were significantly enriched in categories of cell death, apoptosis, migration, etc., many of these genes have been documented to play important roles during tumorigenesis (see Supplementary Data 1). Interestingly, TAGs from CGC and TAG databases, displayed substantial different levels of mRNA expression between the Tibetan and Min pigs (Supplementary Fig. 1b).

Interestingly, we also found that age-standardized death rates for various human cancers including lung, colorectal, liver and breast cancers are significantly lower in Tibet than that in all the other provinces (Supplementary Fig. 1c, d). This analysis was done using the newly updated epidemiological data for China at the provincial level[27], which is similar to the longevity and cancer resistance phenotypes observed in naked and blind mole rats living under extreme hypoxic conditions[11,12]. The above evidence makes it possible to identify putative cancer biomarkers from genes selected for hypoxia adaptation. For this purpose, we examined the function of positively selected genes in domestic mammals from Tibet. Except for *TEX2* and *YTHDF1*, most of the top two ranked positively selected genes

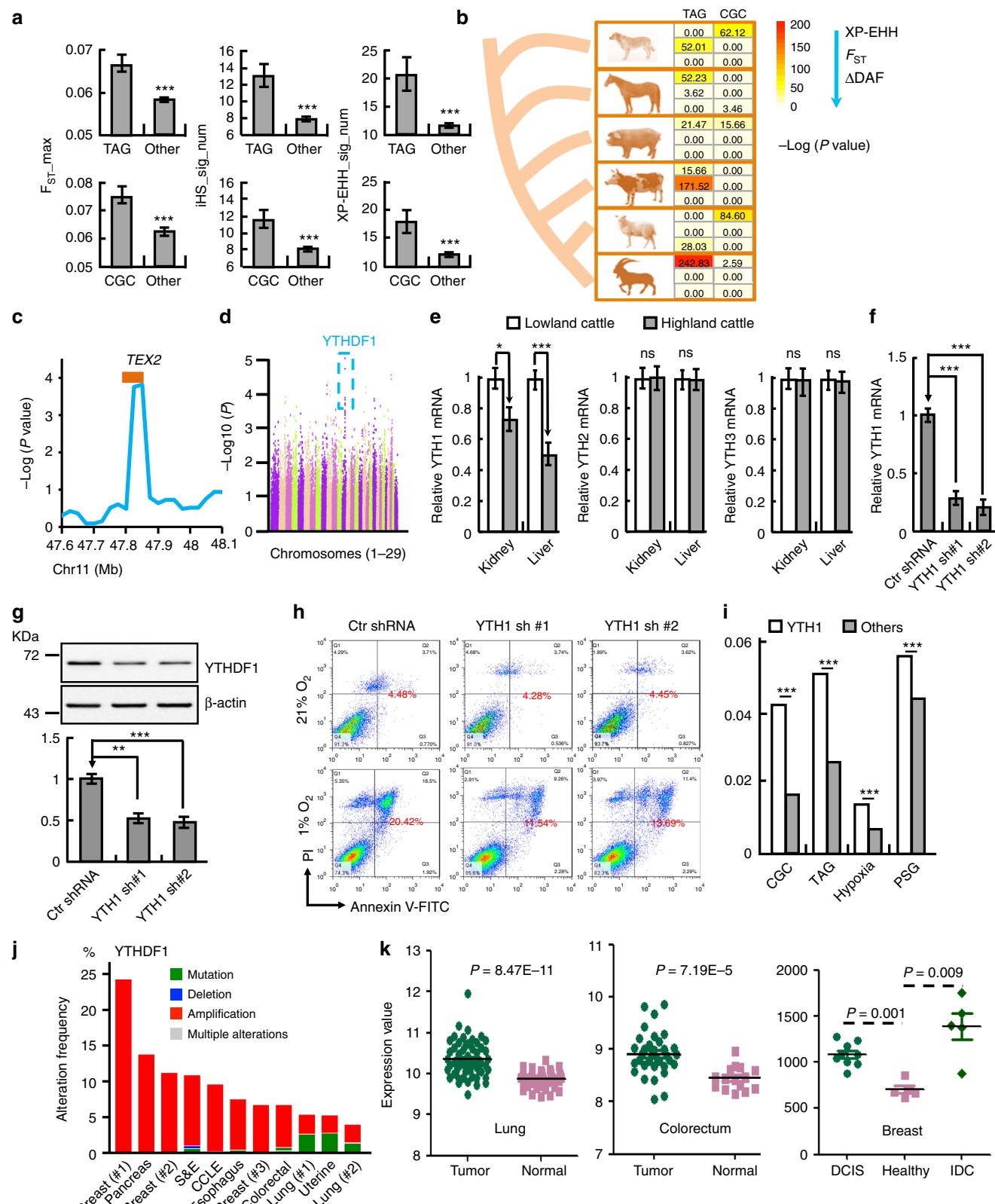

from individual domestic animals, *EPAS1* and *HBB* in dog, *AKAP13* and *PAPSS2* in horse, *IFNLR1* and *MBD3* in pig, *BIRC7* and *YTHDF1* in cattle, *TEX2* and *IKZF1* in sheep, *DSG3* and *NOL4* in goat, have been documented to play pivotal roles in different cancer types (Fig. 1c, d; see Supplementary Data 2). This result strongly supports the feasibility of this approach,

and suggests that *YTHDF1* and *TEX2* are likely candidate genes that play important roles in cancer progression.

**YTHDF1 in hypoxia adaptation and cancer progression**. Due to the frequent decreased expression of *TEX2* in various cancers and lack of documented functions (Supplementary Fig. 1e), we

**Fig. 1** YTHDF1 is a hypoxia adaptation gene. **a** The signature of positive selection on cancer related genes in Tibetan humans evaluated by different statistics. For each gene, $F_{ST}$\_max is the maximum $F_{ST}$ value among all SNPs within the gene. iHS_sig_num and XP-EHH_sig_num for each gene are the numbers of SNPs exhibiting significant high values of iHS (integrated haplotype score) and XP-EHH (cross population extended haplotype homozygosity) respectively. Mann–Whitney U tests were used to test the statistical significances. TAG, tumor-associated genes are from www.binfo.ncku.edu.tw/TAG/; CGC, refers to Cancer Gene Census database. Other, refers to genome wide other genes. **b** Significantly higher values of XP-EHH, $F_{ST}$ and ΔDAF (the difference of the derived allele frequencies) were found for SNPs in tumor related genes than in other genome wide genes. The values of two columns represent the –log10 transformed P-values calculated using Mann–Whitney U tests. **c** Positive selection of TEX2 in Tibetan sheep. Sliding window analysis (size: 50 kb, step: 25 kb) was performed with -log 10 (empirical P value). **d** The genomic landscape of the signature of positive selection in the highland cattle genome. Sliding window analysis (size: 50 kb, step: 25 kb) was performed with -log 10 (empirical P value) for autosome 1 to 29. **e** The mRNA expression of YTHDF1, but not YTHDF2 or YTHDF3 is decreased in highland cattle. **f, g** Validating the efficiency of shRNAs targeting to *YTHDF1* by both real-time RT-PCR (**f**) and western blot (**g**). **h**, Suppression of cellular apoptosis by depleting YTHDF1 under 1% $O_2$ hypoxic condition. **i** YTHDF1 interacting $m^6$-mRNA transcripts overlapped more with CGC, TAG, Hypoxia response genes and PSG (positive selected genes). **j** YTHDF1 is frequently amplified in various cancers. Mutation (green), deletion (blue), amplification (red), multiple alterations (gray). The related database was indicated in Supplementary Table 1. **k** Significant differential expression of YTHDF1 between tumor and normal tissues from lung (GEO accession code: GSE10072), colorectum (GSE24514) and breast (GSE21422) cancers. DCIS: ductal carcinoma in situ; IDC: invasive ductal carcinoma. Means ± SEM, *$P < 0.05$; **$P < 0.01$; ***$P < 0.001$, t-test. Ctr = Control shRNA or scrambled shRNA; YTH1 = YTHDF1; YTH2 = YTHDF2; YTH3 = YTHDF3

decided to further corroborate our hypothesis on the roles of YTHDF1, one of the $m^6$A-specific mRNA binding and translation regulating proteins, in hypoxia tolerance and cancer progression[28,29]. Since no amino acid change within YTHDF1 was identified in highland cattle (data not shown), we reasoned that a change in mRNA expression might have occurred during evolution. Indeed, we found that the mRNA expression levels of YTHDF1, but not the other two YTH domain family members YTHDF2 and YTHDF3, were lower in the kidney and liver tissues derived from highland cattle than those from lowland cattle (Fig. 1e). To examine whether the low expression of YTHDF1 correlates with hypoxia adaptation in vitro, we knocked down YTHDF1 mRNA expression in normal human bronchial epithelium cells (BEAS-2B) with 2 independent shRNAs, and indeed found that deficiency of YTHDF1 abrogated hypoxia-induced cellular apoptosis significantly, as examined by Annexin V staining and western blot with PARP and cleaved caspase -3 (CC3) antibodies (Fig. 1f–h, Supplementary Fig. 1f, g).

In addition, we found that YTHDF1 targeting of $m^6$A-mRNA transcripts overlapped more significantly with CGC, TAG, hypoxia related and positive selected genes compared with the rest of the untargeted genes[29], (Fig. 1i), which led us to explore the potential function of YTHDF1 in cancers. We first examined its expression pattern using the TCGA database and the cBioPortal web resource[30], and found that YTHDF1, like KRAS, is frequently mutated and amplified in various cancers (Fig. 1j, k, Supplementary Fig. 1h; Supplementary Table 1), including breast, pancreas, colon, and lung cancers. In contrast, another $m^6$A-modified mRNA reader protein YTHDF2, which recognizes $m^6$A and reduces the stability of its targeted transcripts, is mostly deleted in human cancers (Supplementary Fig. 1h). Because hypoxia-driven molecular event changes have been well established to be able to drive drug resistance, enhance epithelial-to-mesenchymal transition, remodel the extracellular matrix, support cancer stem cells, and facilitate evasion from immune surveillance in NSCLC and other hypoxic tumors[31], we then decided to focus on the functional roles of YTHDF1 in NSCLC. Consistent with web resource databases, we observed that both the protein and mRNA expressions are more prominent in NSCLC cancerous tissues and cell lines (H1975, A549, H838, H1299, GLC-82, SPC-A1 and H1650), compared to paracancerous tissues or normal BEAS-2B cells, respectively (Fig. 2a–d, Supplementary Fig. 2a).

**YTHDF1 regulates NSCLC cell proliferation**. To study gain or loss of function of YTHDF1, we forced expression of human

Flag-tagged YTHDF1 with a lenti-viral system and abrogated the YTHDF1 expression with two independent lenti-viral shRNAs targeted to the 3'-UTR of YTHDF1 mRNA (Materials and Methods). Although we did not detect a significant oncogenic effect after YTHDF1 overexpression (Data not shown), we did observe that knockdown of YTHDF1 inhibits cell proliferation and colony formation in vitro compared to the scrambled shRNA control, which could be rescued by YTHDF1 overexpression (Fig. 2e–h, Supplementary Fig. 2b, c). As YTHDF1 affects ribosome occupancy and translation of $m^6$A-modified mRNAs[29], we performed a proteome-wide screening experiment in YTHDF1 knockdown H1299 cell lines using a tandem mass tags (TMT)-based quantitative proteomic approach to reveal protein targets of YTHDF1. A total of 6986 proteins were identified and quantified in the proteomic experiment, and 1363 proteins were significantly changed in YTHDF1 knockdown cells (fold change > = 1.2, students' $t$ test $P$ value < 0.05) (see Supplementary Data 3). Intriguingly, the two YTHDF1 shRNA groups and the control group were classified into two separate clusters, and 12 pathways were enriched by KEGG (Kyoto Encyclopedia of Genes and Genomes) analysis (Fig. 2i, Supplementary Fig. 2e, f). In addition, we found that many cell cycle checkpoint regulators which have been indicated to play important roles during G0/G1 cell cycle transition[32] including CDK2, CDK4 and cyclin D1 are down-regulated (see Supplementary Data 3). To examine whether YTHDF1 regulates cell cycle transition, we performed flow cytometric analysis which revealed that YTHDF1 abrogation led to a significant increased G0/G1 cells and more p27 protein expression (one of the cyclin-dependent protein kinase inhibitors[33]), than in control cells (Fig. 2j, m, Supplementary Fig. 2h). We also examined DNA synthesis using Click-iT EdU Alexa Fluor Imaging, and found that YTHDF1 ablation in A549 and H1299 cells markedly decreased the ratio of EdU-positive cells. Consistent with the proteomic data (with the exception of CDK6), other G0/G1 cell cycle transition key regulators including CDK2, CDK4, and cyclin D1 proteins were consistently reduced in YTHDF1 knockdown cells, with no observed differential mRNA expression (Fig. 2k–m, Supplementary Fig. 2d–g). In addition to perform the $m^6$A-seq in A549 cells, we sequenced RNA obtained from the immuno-purified complex of YTHDF1 (RIP-seq) to reveal YTHDF1 bound mRNAs, 3,676 genes were shared ($m^6$A-seq + RIP-seq) as high-confident targets of YTHDF1 (Fig. 2n, see Supplementary Data 4), which were mapped to cell cycle and tumor (including lung cancer) related signaling pathways in the KEGG (Kyoto Encyclopedia of Genes and Genomes) pathway database (Fig. 2o, see Supplementary Data 4), and $m^6$A peaks as well as YTHDF1 binding enrichment

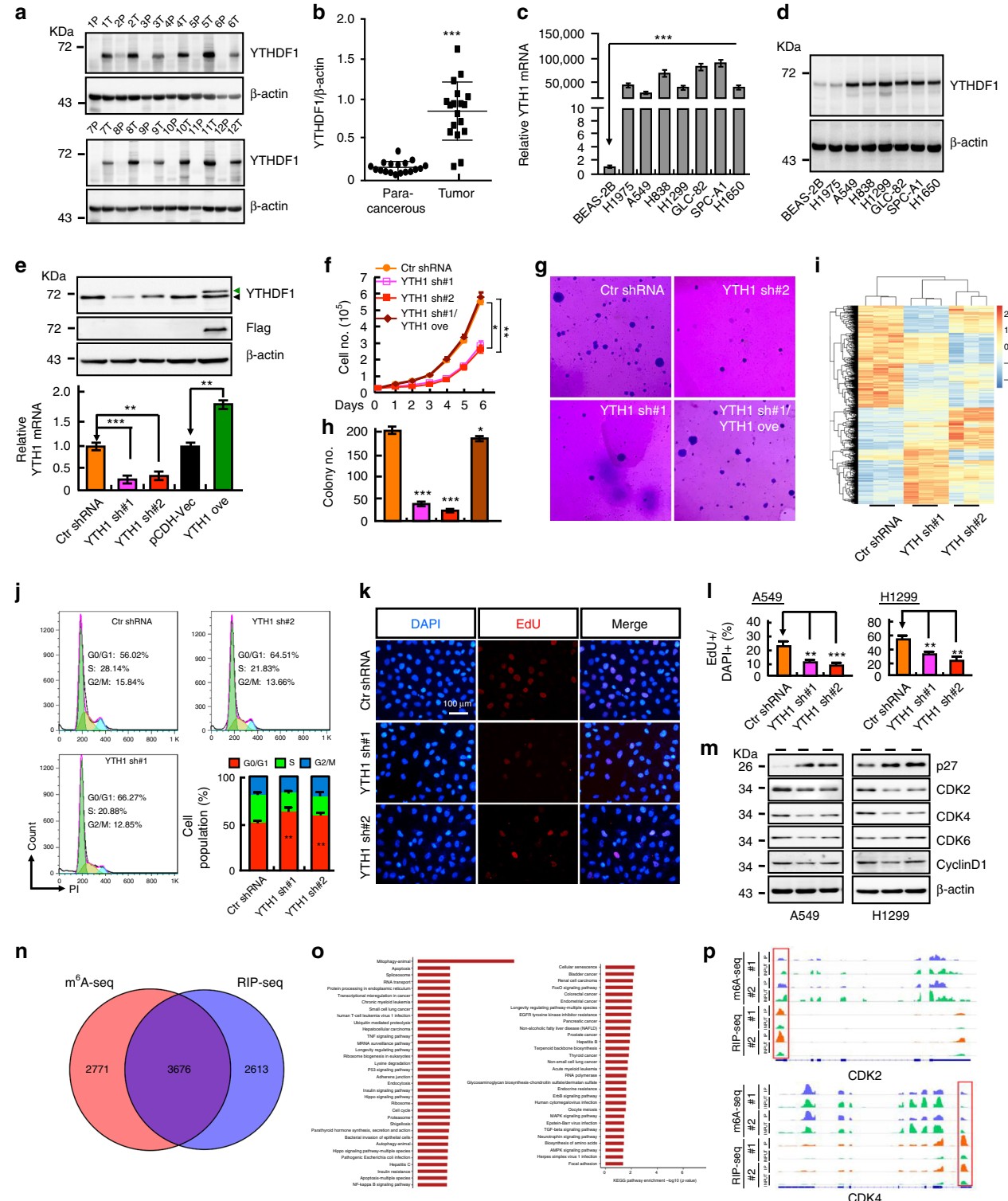

were observed in CDK2 and CDK4 as shown by Integrative Genomics Viewer (IGV) software (Fig. 2p). Furthermore, we showed that knockdown of YTHDF1 in A549 cells significantly reduced CDK2 and cyclin D1 mRNAs in the translating pool (Supplementary Fig. 2i).

**YTHDF1 deficiency inhibits lung cancer progression in vivo.**
We used xenograft tumor formation to examine the in vivo function of YTHDF1. Five-week old male nude mice were randomly divided into indicated groups and injected with cell lines

stably expressing scrambled control shRNA or shRNAs targeting YTHDF1 ($1 \times 10^6$ cells/point subcutaneously). Mice were monitored each day and tumor growth was measured every 3 days. As expected, we observed subcutaneous tumors in the scrambled shRNA control group. Knockdown of YTHDF1 dramatically retarded tumor formation, tumor weights, and volumes compared to scrambled shRNA-transformed cells (Fig. 3a–d).

To understand whether YTHDF1 regulates lung tumor initiation in vivo, we took advantage of YTHDF1$^{lox/lox}$ (Y) mice which were crossed with Kras$^{G12D}$, with or without Trp53$^{lox/lox}$

**Fig. 2** YTHDF1 is elevated in various cancers. **a, b** YTHDF1 is increased in NSCLC cancerous tissues examined by western blot. (**b**) is the quantification data for (**a**). p paracancerous tissue, T tumor. The numbers indicated different tissues. **c, d** Both YTHDF1 mRNA (**c**) and protein (**d**) were increased in various NSCLC cell lines: H1975, A549, H838, H1299, GLC-82, SPC-A1, and H1650, compared with BEAS-2B. **e** Establishment of YTHDF1 overexpression and knockdown A549 cell lines, verified by western blot (top) and Real-time RT-PCR (bottom). Green arrow: exogenous YTHDF1-Flag; black arrow: Endogenous YTHDF1. **f, h** YTHDF1 Knockdown dramatically inhibits A549 cell proliferation (**f**) and colony formation ability (**g**) in vitro. (**h**) is quantification data for (**g**). **i** A heat map of the most significant 1363 altered protein intensities was generated using hierarchical clustering analysis. **j** Effect of YTHDF1 knockdown on the G0/G1 cell population in A549 cells, as measured by PI staining and flow cytometry. Quantification data is also indicated. **k** Knockdown of YTHDF1 significantly decreases DNA synthesis in A549 cells, as measured with a Click-iT EdU Alexa Fluor Imaging Kit. Scale bar: 100μM. **l** Quantification data for the EdU assays in A549 and H1299 cell lines. **m** Effect of YTHDF1 knockdown on protein levels of the G0/G1 cell cycle regulators, including p27, CDK2, CDK4, CDK6, and cyclin D1. Indicated cell extracts were probed with indicated antibodies. **n** Overlap of m$^6$A-seq peaks in A549 cells with RIP-seq for YTHDF1. **o** Significantly enriched KEGG (Kyoto Encyclopedia of Genes and Genomes) pathways for 3676 overlapped genes (p < 0.05). **p** IGV tracks displaying m$^6$A peaks and YTHDF1 binding enrichment in indicated genes from m$^6$A-seq and YTHDF1 RIP-seq in A549 cells, respectively. Blue indicated m$^6$A-seq IP, orange indicated YTHDF1 RIP-seq IP, and green indicated INPUT. Both the m$^6$A-seq and RIP-seq have two replicates. Means ± SEM, *P < 0.05; **P < 0.01; ***P < 0.001; t-test

(KP or K) lung ADC mouse models[34], to generate KPY or KY mice (Supplementary Fig. 3a, b). Mice were then given Adeno-Cre virus by nasal inhalation and sacrificed for gross inspection and pathologic studies at 12 weeks (for KP and KPY mice) or 22 weeks (for K or KY mice) post viral administration (Fig. 3e). We observed that ADC derived from KP or K mice exhibited high YTHDF1 expression compared with the loss of YTHDF1 detection in KY or KPY mice (Fig. 3f, Supplementary Fig. 3d). As compared with reciprocal control KP or K mutant mice, KPY or KY mice showed a dramatic decrease in tumor burden as measured by tumor number and tumor size (large tumors, ≥1 mm$^2$), respectively, indicating that YTHDF1 promotes the lung tumor progression driven by KRAS with or without Trp53 mutation (Fig. 3g, h, Supplementary Fig. 3e, f). Consistently, a lower proliferation rate evidenced by Ki67 staining and an increase of CC3-positive cells were detected in lung tumors from KPY (or KY) mice compared to the KP (or K) mice, suggesting that YTHDF1 deletion had an inhibitory effect on tumor cell growth in vivo (Fig. 3j, k, Supplementary Fig. 3g, h). Detailed pathologic analysis revealed that the majority of the lesions from the KPY mice were lung ADC, indicated by positive immunostaining of both Napsin A and TTF-1 (Fig. 3i).

**YTHDF1 positively correlates with clinical outcome.** Prompted by the results that significant increased YTHDF1 expression was observed in NSCLC tissues and cancerous cell lines (Fig. 2, Supplementary Fig. 2), we further surveyed the protein expression and cellular location of YTHDF1 in NSCLC (including lung SCC and ADC) and noncancerous control lung tissues (NCLT) by immunohistochemical staining (IHC) using tissue microarray. The percentage of positive YTHDF1 expression was significantly higher in NSCLC tissues (55.9%; 272/487) than that in NCLT tissues (42.1%; 64/152) (Fig. 4a, b, Supplementary Fig. 4a; Supplementary Table 2). Surprisingly, when the association between YTHDF1 protein levels and overall survival were analyzed, we noticed that YTHDF1 high expression patients had a better clinical outcome, while low protein expression patients had a significantly adverse outcome (Fig. 4c). Consistent with this, we obtained similar results by applying the Affymetrix gene expression dataset from 1926 lung cancer patients (Fig. 4d)[35].

To explore the underlying mechanism by which low expression of YTHDF1 causes a worse survival rate, we reanalyzed the tissue microarray database used for IHC, and found that 462 NSCLC patients were treated by platinum based chemo-(441/462) or radio-(21/462) therapy alone, whereas 25 patients were treated by both chemo- and radio-therapies (Supplementary Table 2). Since cisplatin (cis-Diamminedichloroplatinum, DDP) was one of the most preferred first line drugs used in these patients, we then proposed that high or low expression of YTHDF1 might sensitize

or restrain, respectively, the cancerous cells responding to DDP after surgery, which in turn results in either a better or worse overall survival rate. To test this hypothesis in vivo, KP and KPY mice were sacrificed for gross inspection and pathologic studies 12 weeks after they were given Adeno-Cre virus by nasal inhalation for 8 weeks, vehicle Phosphate-Buffered Saline (PBS), or were injected with DDP (7 mg/kg) by intraperitoneal injection every week (Fig. 4e). Histological examination revealed that the tumor burden was dramatically repressed in KP mice comparing with KPY mice after DDP treatment, evidenced by dramatic reduction of tumor number, tumor size and Ki67 positive cell numbers, accompanied with increased CC3-positive immunostainings (Fig. 4f, Supplementary Fig. 4h). Furthermore, similar phenotypes were validated by xenografts in mice. YTHDF1 knockdown with control cells were grown as xenografts in nude mice. After tumors reached about 50 mm$^3$ in size, mice were randomized for treatment with PBS or DDP. For A549 xenografts, control tumors treated with PBS grew to average 470, 253 and 145 mm$^3$ in control shRNA, YTHDF1 shRNA #1 and shRNA#2 groups, respectively, 21 days following randomization (Fig. 4g). Interestingly, scramble shRNA tumors treated with DDP grew to ~23% of PBS treated tumor size, however, YTHDF1 knockdown tumors did not show significant difference in tumor size comparing PBS with DDP treatment groups. (Fig. 4g–i, Supplementary Fig. 4b–e). In the DDP treatment tumor groups, dramatically higher proliferation as measured by Ki67 IHC, and lower overall apoptosis indicated by CC3 IHC, in YTHDF1 shRNA tumors were also detected compared with control tumors (Fig. 4h, i, Supplementary Fig. 4f, g). In vitro, we also found that YTHDF1 is downregulated in cisplatin-resistant A549 cells (A549/DDP), whereas forced expression of YTHDF1 in A549/DDP or depleting YTHDF1 in A549 and H1299 cells promoted or inhibited cellular viabilities, respectively (Fig. 5d, e, Supplementary Fig. 4i, j, and 5a, e, f). To corroborate this phenotype, we also treated the NSCLC cancerous cells with radiation or navitoclax, an inhibitor of the anti-apoptotic factors BCL-xL and BCL-2[36]. We found that YTHDF1 knockdown inhibited cellular apoptosis in radiation but not navitoclax treatment group (Supplementary Fig. 5g).

**YTHDF1 functions through Keap1-Nrf2-AKR1C1 axis.** Hypoxia can induce cellular apoptosis through energy deprivation and radical formation including excess ROS generation[37,38]. Low oxygen levels occurring in hypoxic solid tumors is often associated with resistance to radio-, chemo- or targeted-therapies[39–43]. ROS are produced metabolically as byproducts by mitochondria and other cellular elements, as well as by external factors including hypoxia, smoking, pollutants, chemodrugs and radiation[44,45]. NF-E2 p45-related factor 2 (Nrf2) is a

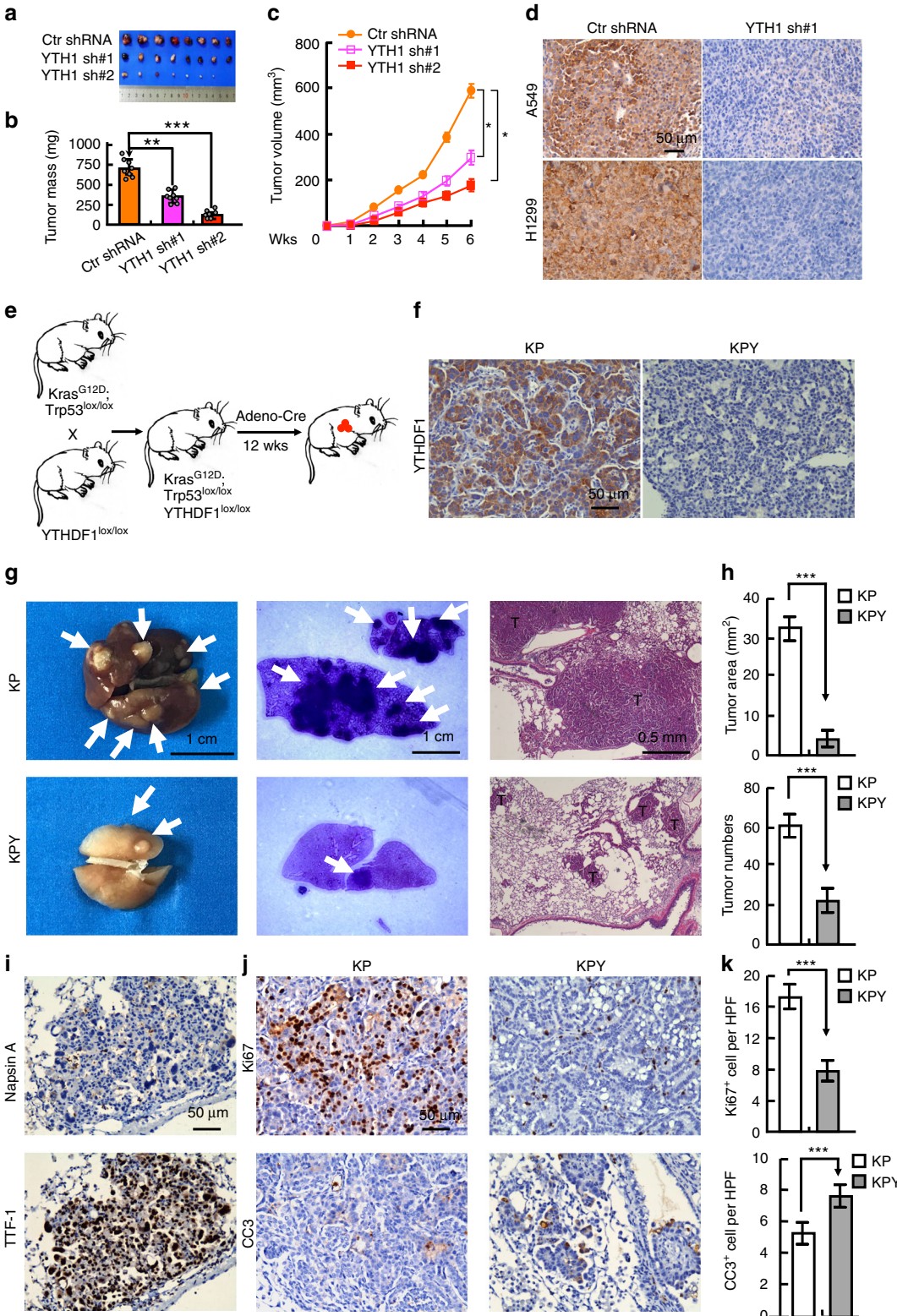

**Fig. 3** YTHDF1 depletion inhibits de novo ADC progression. **a** Xenograft tumor masses harvested from indicated A549 cell lines after tumors had grown for 42 days. **b**, **c** YTHDF1 elimination inhibits xenograft tumor weight (**b**) and volumes (**c**) in male nude mice. **d** Representative IHC images showing YTHDF1 knockdown in xenografts. Scale bar: 50 μm. **e** A scheme of Adeno-Cre treatment-induced KP and KPY mouse models. **f** YTHDF1 is upregulated in ADC from KP mice compared with the complete loss of YTHDF1 in KPY mouse tumors. Scale bar, 50 μm. **g**, **h** Representative images of H&E-stained lung sections of KP or KPY mice at 12 weeks after viral administration (**g**). Quantification data for the tumor numbers and tumor size (≥1 mm$^2$) in KP or KPY mice (**h**). Scale bars are indicated in each image. T Tumor tissues. **i** Representative images of positive staining of Napsin A and TTF-1 for KP mice ADC tumors. Scale bar, 50 μm. **j**, **k** Representative images of Ki67 and CC3 immunostaining (**j**) of lung tumors and statistical analyses of the Ki67- and CC3-positive index in the indicated genotype (**k**), respectively. CC3: cleaved caspase 3. Scale bar, 50 μm. Means ± SEM, *$P < 0.05$; $t$-test

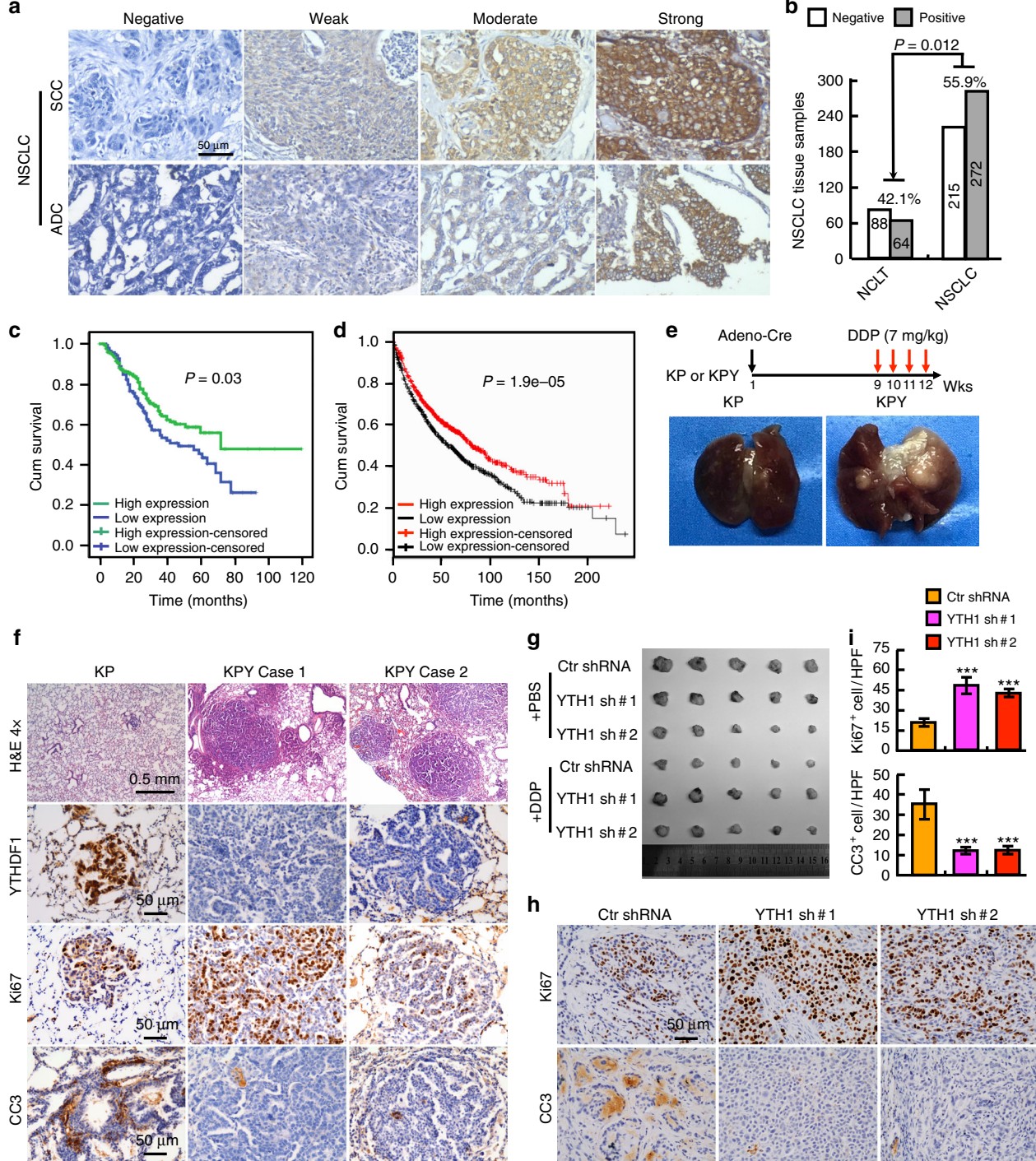

**Fig. 4** YTHDF1 high expression correlates with a better clinical outcome. **a** IHC staining, ×200. The negative, weak, moderate and strong expression of YTHDF1 protein expression patterns in lung squamous cell carcinoma (SCC) or lung adenocarcinoma (ADC). **b** Quantification data for IHC (**a**). **c**, **d** YTHDF1 high expression in NSCLC correlates with better clinical outcome, data generated from IHC staining (**c**) and network database (**d**). **e** Schematic diagram shows the experimental strategy for cisplatin (DDP) treatment in KP and KPY mice. Representative lung tissues after DDP treatment are shown. **f** Representative H&E and immunostaining for tumors collected from KP and KPY mice after DDP treatment. Scale bars are indicated for each image. Antibodies are: YTHDF1, Ki67, CC3. **g** Xenograft tumor masses harvested from indicated mice treated by indicated conditions. DDP: 7 mg per kg, once a week for 3 weeks. **h**, **i** Representative IHC staining for indicated xenograft tumors after DDP treatment. Scale bar: 50 µm. Antibodies are: Ki67 and CC3. (**i**) is the quantification data for (**h**). Means ± SEM, *$P < 0.05$; **$P < 0.01$; ***$P < 0.001$; $t$-test

transcription factor playing a key role in endogenous antioxidant processes, which regulates the expression of antioxidant genes including aldo-keto reductases 1C1 and 1C2, (AKR1C1 and AKR1C2), heme oxygenase 1 (HO-1) and NADP(H), under both physiological and oxidative stress conditions[46]. Keap1 (Kelch-like

ECH-associated protein 1) functions as a negative regulator for Nrf2, which is deactivated under oxidative stress condition[47]. To explore the underlying mechanism, we reanalyzed our mass-spec data and noticed that Keap1 and Nrf2 were inversely regulated upon YTHDF1 knockdown, as shown by decreased Keap1 but

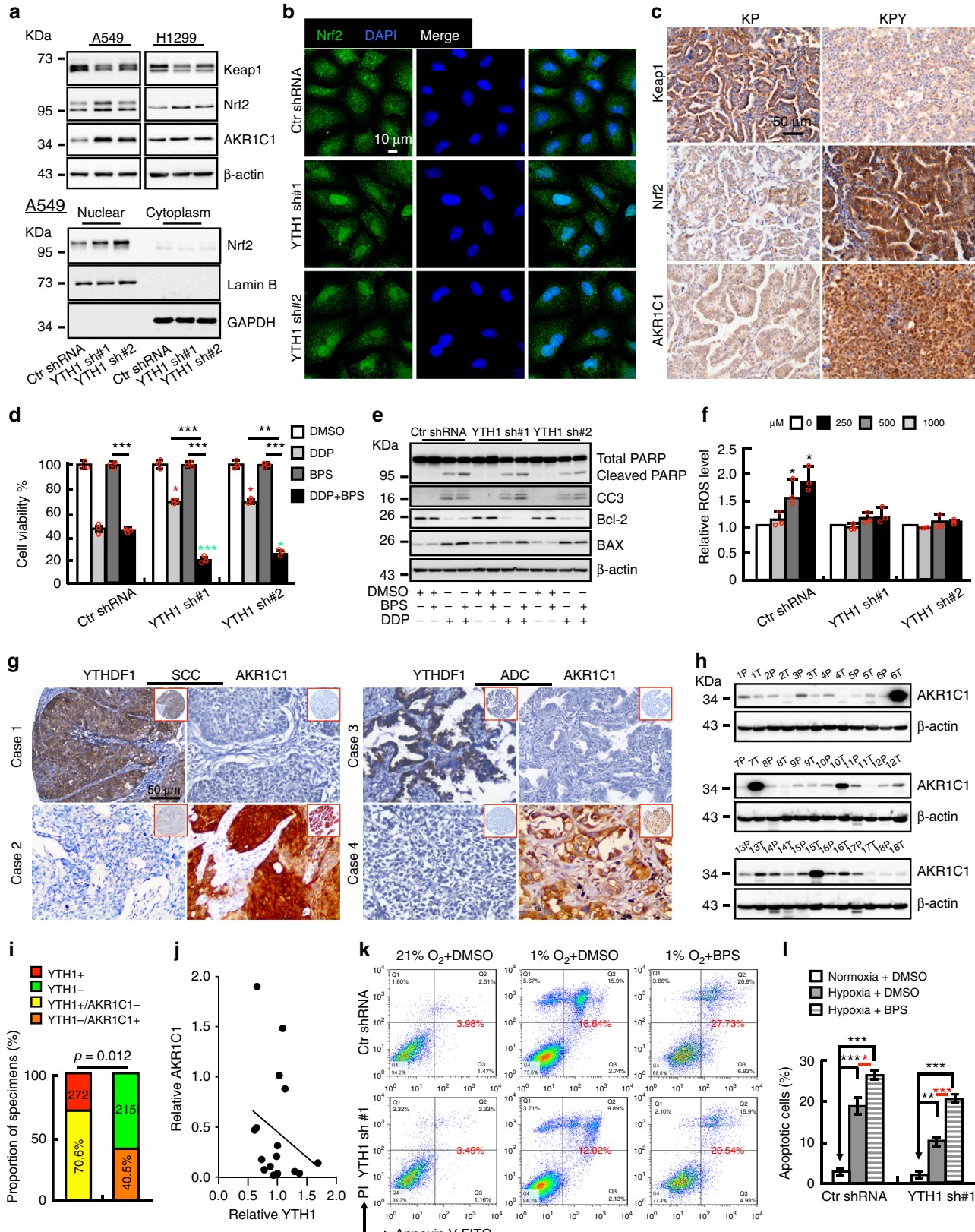

increased Nrf2 and its downstream responding factor AKR1C1 protein expression (see Supplementary Data 3). AKR1C1 belongs to the human aldo-keto reductase (AKR) family, whose aberrant expression has been shown to be induced during development of resistance to a variety of chemotherapeutic drugs with carcinogenesis of NSCLC, breast and ovarian cancers[48,49]. Therefore, we

decided to validate whether the Keap1-Nrf2-AKR1C1 axis is the mediator for the YTHDF1 function responding to hypoxia- or DDP- induced cellular apoptosis. We firstly confirmed that the mRNAs for Keap1 and Nrf2 were not affected upon DDP treatment by Real-time RT-PCR (Supplementary Fig. 5d). By western blot, we further confirmed that Keap1 was decreased whereas

**Fig. 5** The Keap1-Nrf2-AKR1C1 axis is the mediator for YTHDF1. **a** After DDP (15 μM) treatment in A549 cells, indicated cell extracts were examined by western blot. Antibodies: Keap1, Nrf2, AKR1C1, β-actin, Lamin B (Nuclear fraction), GAPDH (cytosol fraction). **b** Representative images indicating Nrf2 nuclear accumulation after YTHDF1 knockdown in A549 cells treated by 15 μM DDP. Scale bar: 10 μm. **c**, Representative IHC stain,ng of KP and KPY tumors for Keap1, Nrf2 and AKR1C1 expressions. Scale bar; 50 μm. **d**, **e** Effect of DDP and/or AKR1C1 inhibitor BPS on cell viabilities of indicated A549 cell lines (**d**), which were further validated by western blot (**e**). DDP: 15 μM, BPS: 40μM. Indicated cells were pretreated by DDP for 12 h followed by BPS treatment. Indicated total extracts were probed with indicated antibodies: PARP, CC3, Bcl-2, BAX and β-actin. Red stars: DDP treatment group comparison; Green stars: DDP + BPS treatment group comparison. **f** YTHDF1 knockdown exhibited antioxidant functions. Indicated A549 cell lines were treated with indicated doses of $H_2O_2$ for 12 h, intracellular ROS levels were measured. Representative histograms of ROS analysis are shown. **g**, **j** YTHDF1 negatively correlates with AKR1C1 expressions in NSCLC (SCC and ADC) tissues, validated by both IHC staining in TMA (**g**) and fresh clinical tissues using western blot (**h**). (**i**) is the quantification data for (**g**). (**j**) is the quantification data for (**h**). **k**, **l** YTHDF1 knockdown dependent resistance to hypoxia-induced cellular apoptosis is reversed by treating cells with 40 μM BPS. After 72 h treatment under hypoxia condition, indicated cells are stained with Annexin V/PI, and the percentage of apoptotic cells was assessed by flow cytometry (**k**). (**l**) is the quantification data for (**k**). Means ± SEM, *$P < 0.05$; **$P < 0.01$; ***$P < 0.001$; $t$-test

Nrf2 and AKR1C1 were increased upon YTHDF1 knockdown after DDP treatment, accompanied by increased nuclear accumulation of Nrf2 protein-one of the markers for the activation of Nrf2 and its downstream targeted genes including *AKRs*, *GR*, *GCLC*, *GCLM*, and *NQO1*[50] (Fig. 5a, b, Supplementary Fig. 5b–d). To further determine whether AKR1C1 mediates the regulation of cellular responses to DDP by YTHDF1, we inhibited AKR1C1 by 3-bromo-5-phenylsalicylic acid (BPS), an inhibitor designed based on the structure of AKR1C1[49], and found that BPS in combination with $IC_{50}$-DDP was significantly more effective than either agent alone in decreasing cellular viability. CC3 and PARP western blot also confirmed this (Fig. 5d, e, Supplementary Fig. 5a, e, f). The protective role of YTHDF1 knockdown against the cellular stress response was verified by measuring intracellular ROS following $H_2O_2$ challenge. While $H_2O_2$ did not induce a measurable amount of ROS in YTHDF1 shRNAs cells, it increased the level of ROS in control shRNA cells in a dose-dependent manner (Fig. 5f). Furthermore, we found that the m6A but not the gene expression level in A549/DDP was significantly different from that in A549 cells, and the m6A modification level of YTHDF1 bound Keap1 transcript was significantly reduced (Supplementary Fig. 5h, i, see Supplementary Data 5). Consistently, we showed that knockdown of YTHDF1 in A549 cells reduced Keap1 mRNAs in the translating pool upon DDP treatment (Supplementary Fig. 5j).

To examine whether this was true in vivo, we performed IHC in KP and KPY mouse ADC tumors, and found that much more Keap1, albeit with moderate Nrf2 and AKR1C1 proteins, were detected in KP mouse tumors, whereas less Keap1 and much Nrf2 and AKR1C1 proteins, both in the cytosol and nuclear fractions, were observed in KPY mouse tumors (Fig. 5c). To confirm whether YTHDF1 was negatively associated with AKR1C1, we performed immunostaining for AKR1C1 in the NSCLC TMA again, and revealed that 70.6% (192/272) of YTHDF1 positive NSCLC tissues were AKR1C1 negative, whereas 40.5% (87/215) YTHDF1 negative NSCLC tissues were AKR1C1 positive, ($p =$ 0.012), and a similar negative correlation between YTHDF1 and AKR1C1 expression was also validated by western blot in the same clinical tissues as used in Fig. 2 (Fig. 5g–j). In addition, the overall survival rate for YTHDF1 negative and AKR1C1 positive NSCLC patients were significant worse than other immuno-type patients (Supplementary Fig 5k). Consistently, we found that YTHDF1 was decreased whereas AKR1C1 was increased by IHC in 100% (6/6) NSCLC patients resistant to platinum based neoadjuvant chemotherapy, whose lung cancer progression were examined by computerized tomography (CT) scan. However the expression pattern was reversed in 75% (3/4) responders (Supplementary Fig 5l, m). Since AKRs family members are also important for hypoxia-induced ROS clearance, we hypothesized that the same mechanism was employed in YTHDF1 deficient BEAS-2B cells resisting to hypoxia-induced cellular apoptosis. As

expected, we found that the decreased cellular apoptosis induced by hypoxia was dramatically increased by treating cells with the AKR1C1 specific inhibitor BPS (Fig. 5k, l).

## Discussion
Unraveling the mechanisms of adaptation to hypoxia will improve our understanding of not only mammalian hibernation, but also unsolved clinical, military, and space travel problems[51]. Previous genome-wide scans have identified many candidate genes including molecules involved in HIF pathway contributing to high-altitude hypoxia adaptation[8]. In addition, HIF pathway has also been found to be involved in various human cancers and offers ideal targets for small molecule intervention[39,52–54]. However, a hostile hypoxic environment has been shown to either promote or inhibit cancer progression[10,55]. Here, we have shown that TAGs evolved rapidly in Tibetans and Tibetan domestic mammals. Although the molecular events important for both hypoxia adaptation and hypoxic solid tumors may behave differently, evolutionary studies using hypoxia adapted animals represent an alternative source to identify molecular events important for cancer progression[11,12]. Here, using the large-scale population genome and transcriptome data of Tibetan domestic mammals, we have identified YTHDF1-an N6-methyladenosine (m6A)-specific RNA binding protein, whose low expression is critical for highland cattle hypoxia adaptation and frequent amplification is identified in many hypoxic solid tumors. Interestingly, Shi et al. showed that m6A facilitates hippocampus-dependent learning and memory through YTHDF1, indicating the important roles of YTHDF1 during development[56]. Recent finding also showed that the durable neoantigen-specific immunity is regulated by YTHDF1, and YTHDF1 deficient mice showed an elevated antigen-specific CD8 + T cell antitumor response[57]. Importantly, we found that YTHDF1 regulates NSCLC cancerous tissues or cells responding to DDP-dependent chemotherapy, indicating that understanding the mechanisms of resistance to hypoxia-induced apoptosis might lead to more specific treatments for hypoxic solid tumors.

Since lung cancer is the leading cause of cancer related death worldwide, and NSCLC accounts for approximately 85% of all cases, we further characterized the functional roles of YTHDF1 in NSCLC[58]. As proposed in our model (Fig. 6), under normoxia conditions, constitutive activation of CDK-cyclin complexes, induced by YTHDF1 amplification, may contribute not only to uncontrolled cell proliferation but also to genomic and chromosomal instability, resulting in cancer progression. Therefore, it is promising to suggest increasing the overall survival of NSCLC patients by repressing YTHDF1. However, the clinical correlation analysis shows the adverse result. Importantly, YTHDF1 deficiency caused resistance of DDP or hypoxia-induced cellular apoptosis in cancerous and normal cells, respectively. The anticancer mechanisms of DDP are caused not only by its

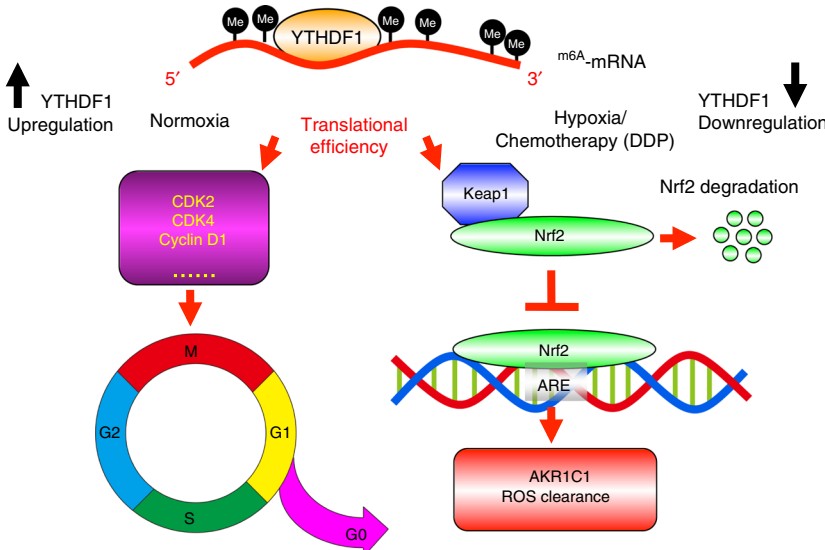

**Fig. 6** A working model for YTHDF1. As proposed in this model, under normoxia conditions for NSCLC progression, YTHDF1 amplification or high expression mainly promotes the translational efficiency of targeted m6A-modified transcripts, such as cell cycle regulators including CDK2, CDK4, cyclin D1, etc., in cancerous cells, which causes uncontrolled cancerous cell proliferation. On the other hand, when YTHDF1 low expression NSCLC patients encounter chemotherapy stress condition, especially cisplatin treatment-induced reactive oxygen species (ROS) accumulation, decreased Keap1 translational efficiency leads to upregulation of Nrf2 and its downstream ARE responding factor antioxidant AKR1C1, which in turn renders cancerous cells resistance to cisplatin treatment, and eventually results in a worse clinical outcome. In contrast to YTHDF1 high expression in NSCLC cancerous tissues, YTHDF1 is decreased in highland cattle compared to that in lowland cattle. The Keap1-Nrf2-AKR1C1 axis is the same mediator for YTHDF1 low expression dependent hypoxia adaptation. Thus, the balance of YTHDF1 expression and its targeted m6A-modified mRNAs or interacting proteins between normoxia and hypoxia/cisplatin treatment (stressful) conditions is important both for non-pathological homeostasis and various human cancers, and it will be necessary to characterize these molecular events in the future

covalent binding with DNA, but also by ROS formation resulting into activation of p53 and caspases[59]. Mechanistic studies indicate that ablation of YTHDF1 decreases translational efficiency of the m6A-modified Keap1 transcript in an oxidative stress state induced by DDP, which in turn activates the antioxidant ROS clearance system (Nrf2-AKR1C1) leading to DDP resistance and a worse clinical outcome for NSCLC patients after platinum based chemotherapy.

In contrast to YTHDF1 increased expression in NSCLC cancerous tissues, YTHDF1 is decreased in highland cattle compared to that in their lowland counterparts, which also results in increased expression of AKR1C1 and a better antioxidant defense response under hypoxic stress conditions. Thus, the balance of YTHDF1 expression and its targeted m6A-modified mRNAs or interacting proteins between normoxia and hypoxia conditions is important both for non-pathological homeostasis and various human cancers. These findings may have important ramifications on targeting YTHDF1 for treating NSCLC, and it will be necessary to distinguish the specific YTHDF1 substrates respectively in the future. Consistent with our findings, YTHDF1 protein is decreased whereas AKR1C1 is increased in platinum based neoadjuvant chemotherapy non-responders. Although further pre-clinical or clinical studies and statistical analysis are needed, our findings not only provide a route to reduce the cyto-toxicity of DDP during neoadjuvant or adjuvant chemotherapy[59], but also provide a potential therapeutic strategy to improve the clinical outcome of YTHDF1 low expressing NSCLC patients, by combining usage of AKR1C1 specific (pre-) clinical inhibitors with platinum based chemotherapy in future.

$N^6$-methyladenosine (m6A), being the most profound modification in mRNA of many eukaryotic species, plays a pivotal role in various bioprocesses including tissue development, self-renewal and differentiation of stem cells, DNA damage response and cancers. Multiple writers, erasers or readers of m6A

have been demonstrated to play important roles in various cancers (e.g., Lung caner, breast cancer, endometrial cancer, glioma, HCC and acute myeloid leukemia)[20,60–62]. For example, Methyltransferase-like 3 (METTL3), one of the m6A writers, is upregulated in lung adenocarcinoma and play oncogenic effect by promoting translation of its target mRNA transcripts including *EGFR* and *TAZ*[63]. Overexpression of m6A demethylase ALKBH5 was associated with lower survival time of glioma[64]. Currently, it is not clear why YTHDF1 is downregulated in highlanders but upregulated in hypoxic solid tumors, and whether or not YTHDF1 interplays with other m6A modifiers (writers, erasers and/or readers) in NSCLC. Different from our observation that YTHDF1 low expression correlates with worse NSCLC clinical outcome by IHC analysis using tissue microarray, another study using TCGA RNA-seq data showed that YTHDF1 was upregulated in hepatocellular carcinoma (HCC), which correlated with significant shorter OS and DFS survival rate[65]. Another study also showed that oncogene c-Myc promotes YTHDF1 expression in colorectal cancer, whose high expression was associated with poorer overall survival[66]. The above apparent discrepancies suggest that we need to take both the protein and mRNA expressions of YTHDF1 into consideration to characterize its functional roles in future. In addition, the m6A modified mRNA targets of YTHDF1 could be time- and cellular context- dependent, which leads to the differential functions of YTHDF1 in various cancers progression.

For the last decade lung cancer has been simplistically divided into NSCLC and small-cell lung cancer on the basis of limiting molecular subsets[67]. Therefore, identification of additional molecular events is essential for developing therapies against NSCLC. Our findings show that YTHDF1 is dramatically upregulated in K or KP mice de novo ADC tumors compared with paracancerous tissues. YTHDF1 deficiency renders tumors resistant to cisplatin dependent chemotherapy in vivo, suggesting that *KRAS* mutations

along with YTHDF1 mutations, or expression level, should be simultaneously considered to predict the efficacy of NSCLC therapies. Furthermore, the physiological role of YTHDF1 for the development of lung and other tissues needs to be deciphered to distinguish the differential effect of specific m6A modifications during development and tumorigenesis.

Together, our findings represent the application of evolutionary theory, using genomic and transcriptome data from hypoxia adapted Tibet domestic mammals, to identify a ranked list of genes that can form the basis for not only hypoxia adaptation of species to high elevations, but also have a broad impact in cancer biology and therapy. Finally, we advocate for the requirement of evolutionary methods and concepts to understand cancer progression and the hallmarks of cancer from different angle[68].

## Methods

**Human SNPs data.** Human genome wide SNPs, genotyped by the Affymetrix Genome-Wide Human SNP 6.0 Array, from 31 unrelated Tibetans were downloaded from a previous study[22] and compared with genotype data for Han Chinese from HapMap (phase II, http://hapmap.ncbi.nlm.nih.gov/). We also used whole genome sequences of Han and Tibetan humans from a previous study[24,25]. The iHS, XP-EHH and $F_{ST}$ values were calculated for each SNP to evaluate the evolution of Tibetans.

**SNP data and positively selected genes.** Transcriptomic data of lung tissues of four Tibetan pigs and four Min pigs were from our other study[69]. Briefly, a total of 134 genomes from 19 Tibetan (highland) goats, 24 Tibetan horses, 20 Tibetan sheep, 11 Tibetan dogs, 19 Tibetan cattle and 41 Tibetan pigs, together with 193 genomes from 20 goat, 10 horse, 70 sheep, 34 dogs, 30 cattle, and 29 pigs from the lowlands were obtained for comparative population genomic analysis. SNPs for each species were called. Population differentiation ($F_{ST}$) between domestic mammals in the highland and lowland was calculated for each SNP[23]. The XP-EHH value for each SNP was calculated by the XP-EHH program (http://hgdp. uchicago.edu/Software/). ΔDAF (the difference of derived allele frequencies) was calculated for each SNP as the derived allele frequency (DAF) in the domestic animals in highland minus the DAF in domestic animals from the lowland. $F_{ST}$, XP-EHH, and ΔDAF (the difference of the derived allele frequencies) were integrated to identify potential positive selection in the Tibetan domestic mammals: $iFXD = \prod_{i=1}^{n} \frac{Psi}{1-Psi}$, where where $i$ is different methods and $Ps$ represent the probability under positive selection. The $iFXD$ value of each protein encoding gene was calculated by averaging the $iFXD$ values of all SNPs within a gene. The top two genes showing highest level $iFXD$ values were described in the main text.

**Genes associated with cancer.** Genes whose mutations have been causally implicated in cancer, were obtained from the CGC database[21] (http://cancer.sanger. ac.uk/cancergenome/projects/census/). Other TAGs were downloaded from (http://www.binfo.ncku.edu.tw/TAG/). One-to-one orthologous genes between human and domestic mammals including dog, horse, pig, cattle, and sheep were downloaded from Ensembl (version 72) by BioMart. One-to-one orthologous genes between human and goat were retrieved by reciprocal best-hit BLASTP search.

**Transcriptomic analysis of lung tissues of Tibetan pigs.** Transcriptomic data of lung tissues of four Tibetan pigs and four Min pigs were from our other study[69]. The trimmed reads were aligned against the reference genome of *Sus scrofa* (Sscrfa 10.2) using TopHat v2.0.4 with default parameters[70] and using the gene annotation available at Ensembl v77. Transcripts were initially assembled using the reference annotation based transcript assembly method in Cufflinks[71]. Newly generated transcripts were subsequently extracted from the output of the assemblers using a custom Perl script. The reference annotation was then merged with the newly generated transcripts to generate another reference annotation file. To quantify gene expression, the FPKM (Fragments Per Kilobase of transcript per Million mapped reads) values were calculated using Cuffmerge without RABT. Finally, CuffDiff was applied to identify differentially expressed genes[71]. We considered a FDR (false discovery rate) cutoff of <0.01 to identify significantly differentially expressed genes between Tibetan and Min pig breeds.

**Expression data from different types of human cancers.** Expression data of genes YTHDF1 and TEX2 were downloaded from NCBI GEO with accession codes including GSE10072, GSE28735, GSE24514, GSE21422, GSE19804 and GSE9574.

**Mouse colony, mouse treatment, and tumor biology studies.** $Kras^{G12D}$ with or without $Trp53^{lox/lox}$ (KP or K) lung adenocarcinoma (ADC) mouse model was generously provided by Hongbin Ji in SIBS, CAS[34]. Mouse care and treatment was approved by the Animal Care and Use Committee at the Kunming Institute of Zoology, Chinese Academy of Sciences. $YTHDF1^{lox/lox}$ mice were generated using a

CRISPR/Cas9 system in C57BL/6J mouse background by Shanghai Model Organisms Center, Inc (Shanghai, China). The YTHDF1 donor vector containing flox sites flanking Exon3 of YTHDF1 gene was cloned, and 4 sgRNAs targeting to Intron2 and Intron3 were generated in vitro. Two sgRNA target sites for Intron2 were 5′-GCATGTGTCCGCTATTTGCC-3′ and 5′- CCCAAGGTGGGACCGAA CCC-3′. Two sgRNA target sites for Intron3 were 5′-TAATGGTGTATAGGACTGTA-3′ and 5′-CTAGGAGAGTAGGTAAGTTC-3′. The donor vector with two sgRNA and Cas9 mRNA was microinjected into C57BL/6J fertilized eggs. F0 generation mice positive for homologous recombination were identified by PCR and confirmed by sequencing. The primers (P1-P4) used for genotyping the correct homology recombination were P1: 5′- TGTGCCCTTCAACCCAGTG -3′ and P2: 5′-CTCGGTAGCTCCCCAGTATCAT-3′ for the correct 5′ homology arm recombination, and P3: 5′- CAGTCCAATCCGGTGAGTTTATCT-3′ and P4: 5′- AAGCTATCCACCTCCCTCTGTATG -3′ for the correct 3′ homology arm recombination. The genotype of F1 generation YTHDF1 flox heterozygous mice were identified by PCR. All mouse protocols were approved by Kunming Institute of Zoology Animal Care and Use Committee.

$YTHDF1^{lox/lox}$ mice were crossed with KP or K mice to generate KPY or KY mice. Mice were treated via nasal inhalation of adenovirus carrying Cre recombinase ($5 \times 10^6$ p.f.u for Ad-Cre, Biowit Inc., Shenzhen, Guangdong), and were then killed at indicated times for gross inspection and histopathological examination. For xenograft tumor growth experiment, male nude mice at 5 weeks of age were divided into indicated groups and injected with indicated cell lines. Tumor sizes in all groups were measured every 3 days for 6 weeks using Vernier calipers (Suzhou, China). For the xenograft Cisplatin treatment assay, day 0 was designed when tumors reached around 50 mm3 in volume. DDP 7 mg/kg or carrier (PBS) was injected i.p. 1 time per mice. All mice were sacrificed at the end of the experiment and tumors were harvested and weighed. Representative images were presented, and all experiments were repeated at least 3 times. *P < 0.05; **P < 0.01; ***P < 0.001; t-test. Detailed information on primers and antibodies used in this study is described in Supplementary Data 6. The China maps in Supplementary Fig. 1 were created by an in house excel program: ChinaDataMap.xls, which can be obtained upon request.

**Constructs, cell culture, and shRNA-lenti-viral infection.** Independent shRNAs against different genes targeting different regions were constructed using a pLKO.1 vector. The 3XFlag C-terminal tagged forms of different overexpression genes were synthesized and cloned into a pCDH-MSCV-E2F-eGFP lenti-viral vector. All of the constructs were verified by sequencing, detailed cloning information can be provided upon request. The lenti-viruses were generated according to the manufacturer's protocol. Briefly, supernatants containing different lenti-viruses generated from HEK-293T cells were collected 48 and 72 h post-transfection. HEK-293T was purchased from ATCC, BEAS-2B was kindly provided by Dr. Hongbin Ji at SIBS, CAS. H1975, A549, NCI-H838, H1299. and NCI-H1650 were purchased from Cobioer, China with STR document, GLC-82, SPC-A1 were gifts from Dr. Yunchao Huang, and A549-DDP was a gift from Dr. Shiyong Sun at Emory University, Atlanta, USA. All cells were cultured in RPMI 1640 medium supplemented with 10% fetal bovine serum (FBS) and 1% penicillin/streptomycin then incubated in a humidified atmosphere with 5% CO2 at 37 °C. Cisplatin was purchased from Sigma (Cat#p4394, USA).

**Nanoflow liquid chromatography tandem mass spectrometry.** The indicated cells were washed three times with cold PBS and lysed in SDT lysis buffer (0.2% SDS (m/v), 100 mM DTT, 100 mM Tris, pH = 7.6). All experiments were performed on an Orbitrap Fusion mass spectrometer with nanoLC easy1200 (Thermo Fisher Scientific). Peptides were loaded on a self-packed column (75 µm × 150 mm, 3 µm ReproSil-Pur C18 beads, 120 Å, Dr. Maisch GmbH, Ammerbuch, Germany) and separated with a 90 min gradient at a flow rate of 300 nL/min. Solvent A was 100% H2O and 0.08% formic acid, solvent B was 80% acetonitrile and 0.08% formic acid. The Orbitrap Fusion was programmed in the data-dependent acquisition mode. An MS1 survey scan of 375–1500 m/z in the Orbitrap at a resolution of 120,000 was collected with an AGC target of 400,000 and maximum injection time of 50 ms. Precursor ions were filtered according to monoisotopic precursor selection, charge state (+2 to +7), and dynamic exclusion (45 s with a ±10 ppm window). Then, the most intense precursors were subjected to HCD fragmentation with a duty cycle of 3 s. The instrument parameters were set as the following: 38% normalized collision energy with 5% stepped collision energy, 50,000 resolution, 100,000 AGC target, 105 ms maximum injection time, 105 Da first mass, 1 m/z isolation width. Raw files were processed by search against the UniProt/SwissProt Huamn database.

**m6A-seq and RIP-seq.** Based on the documented procedure[29], total RNA was extracted using Trizol reagent (Invitrogen, CA, USA) following the manufacturer's procedure. The total RNA quality and quantity were analysis of Bioanalyzer 2100 and RNA 6000 Nano LabChip Kit (Agilent, CA, USA) with RIN number >7.0. Approximately more than 50 µg of total RNA was subjected to isolate Poly (A) mRNA with poly-T oligo attached magnetic beads (Invitrogen). The cleaved RNA fragments were subjected to incubated for 2 h at 4 °C with m6A-specific antibody (No. 202003, Synaptic Systems, Germany) in IP buffer (50 mM Tris-HCl, 750 mM NaCl

and 0.5% Igepal CA-630) supplemented with BSA. The mixture was then incubated with protein-A beads and eluted with elution buffer ($1 \times$ IP buffer and 6.7 mM m⁶A). Eluted RNA was precipitated by 75% ethanol. Eluted m⁶A-containing fragments (IP) and untreated input control fragments are converted to final cDNA library in accordance with a strand-specific library preparation by dUTP method. Then we performed the paired-end $2 \times 150$ bp sequencing on an Illumina Novaseq™ 6000 platform at the LC-BIO Bio-tech ltd (Hangzhou, China) following the vendor's recommended protocol. For RIP-seq, input mRNAs and IP with 150–200 ng RNA of each sample were used to generate the library using Illumina kit.

**Polysome profiling.** Based on the documented procedure, we started with nine 15-cm dish of indicated cells. Before collection, cycloheximide (100 µg/ml) was added into cell culture media for 5 min. Cells were collected, washed and lysated with lysis buffer: 5 mM Tris-HCl (pH 7.5), 2.5 mM MgCl2, 1.5 mM KCl, 100 µg/ml cyclo-heximide, 2 mM DTT, 0.5% Triton X-100, 0.5% Sodium Deoxycholate, 200U/ml RNase inhibitor, 1x protease inhibitor cocktail (EDTA-free). The cell supernatants were fractioned (total 72 fractions, 0.45 ml per fraction), and then analyzed by NanoDrop (Thermo Fisher Scientific) for OD$_{260}$. Sample from each fraction was subjected to Real-time RT-PCR analysis of relative mRNA expressions of interested genes, including CDK2, CDK4, cyclin D1, and Keap1.

**Cell proliferation, viability, and ROS assays.** For cell proliferation assays, tested cell lines were plated into 12-well plates and the cell numbers were subsequently counted each day. Apoptotic cell numbers were analyzed by flow cytometry. Indicated cells were plated at a density of 10,000 cells/well with 100µl medium into 96-well plate, 24 h later, indicated drugs were added. Cell viability was evaluated by standard sulforhodamine B(SRB) staining, and each result was validated in triplicate[72]. To examine cellular response to oxidative stress, cells were treated with H$_2$O$_2$ for 12 h, intracellular ROS levels were detected by reactive oxygen species detection assay kit following the manufacturer's protocol (BioVision Catalog #K936–250).

**IHC staining and antibodies.** The IHC staining for samples on the tissue microarrays (TMAs) was carried out using ready-to-use Envision TM + Dual Link System-HRP methods (Dako; Carpintrria, CA). To eliminate nonspecific staining, the slides were incubated with appropriate preimmune serum for 30 min at room temperature. After incubation with a 1:500 dilution of primary antibody (Abcam: ab99080) to YTHDF1 at 4 °C overnight, slides were rinsed with phosphate-buffered saline (PBS) and incubated with a labeled polymer-HRP which was added according to the manufacturer's instructions and incubated for 30 min. Color reaction was developed by using 3, 3'-diaminobenzidine tetrachloride (DAB) chromogen solution. All slides were counterstained with hematoxylin. Positive control slides were included in every experiment in addition to the internal positive controls. The matched IgG isotype antibody was used as a negative control. The scores for immunohistochemical staining of TMA sections were characterized independently by two evaluators, at 200 X magnification based on the staining intensity and extent of staining. Staining intensity for YTHDF1 was scored as 0 (negative), 1 (weak), 2 (moderate), and 3 (strong). Staining extent was scored as 0 (0%), 1 (1–25%), 2 (26–50%), 3 (51–75%), and 4 (76–100%). Agreement between the two evaluators was 95%, and all scoring discrepancies were resolved through discussion between the two evaluators. All the $P$ values were based on the two-sided statistical analysis and $P$-value less than 0.05 was considered to be statistically significant. Uncropped scans of the most important western blots are shown in Supplementary Fig. 6.

**Ethics statement.** Samples were obtained with informed consent and all protocols were approved by The Second Xiangya Hospital of Central South University Ethics Review Board (Scientific and Research Ethics Committee, S-02/2000). Written informed consent was obtained from all patients. Written informed consent was obtained from the next of kin, caretakers, or guardians on the behalf of the minors/children participants involved in this study. Clinical samples from platinum based neoadjuvant chemotherapy resistant NSCLC patients were obtained from the Third Affiliated Hospital of Kunming Medical University and Harbin Medical University Cancer Hospital in China.

**Reporting summary.** Further information on research design is available in the Nature Research Reporting Summary linked to this article.

## Data availability

The datasets generated during and/or analyzed during the current study are available from the corresponding author on reasonable request. Original m6A-seq and RIP-seq data are available: GSE136433. The mass spectrometry proteomics data have been deposited to the ProteomeXchange Consortium via the PRIDE[73] partner repository with the dataset identifier PXD015182.

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

## Acknowledgements

This study was supported jointly and equally by the National Key Research and Development Program of China, Stem Cell and Translational Research (to Y.B.C., 2016YFA0100900) and the Strategic Priority Research Program of the Chinese Academy of Sciences (to Y.B.C. and D.D.W., XDB13000000). This work was also supported by National Nature Science Foundation of China (U1502224, 81772996, 81672764, U1702283, 81773218), and Yunnan Applied Basic Research Projects (2016FA009). Y.B.C was supported by Yunnan Province High-level Talents Introduced Program (2013HA021), and C.P.Y was also supported by the Chinese Academy of Sciences Western Light Program, Youth Innovation Promotion Association, CAS. S.H.X was supported by the National Natural Science Foundation of China (91731303, 31525014) and the UK Royal Society-Newton Advanced Fellowship (NAF/R1/191094).

## Author contributions

Y.B.C., D.D.W. and C.P.Y. developed the hypothesis. D.D.W. supervised the genomic and transcriptome data analysis. Y.B.C. and C.P.Y. led and designed the functional analysis for YTHDF1 in NSCLC, Y.L.S. performed the biochemical experiments, S.Q.F. and M.G.W. performed the TMA analysis. Z.X.Z. and X.Y.L. performed the RIP-seq and m6A-seq data analysis. H.Z. and J.G. performed mass-spec data analysis. L.P.J., Q.S.S., P.F.X., L.Z., Z.Z.Y, J.M.Z., S.H.X., H.B.J. and P.S. performed the in vivo experiments. Y.C.H and Y.C.Z provided fresh NSCLC clinical samples. We thank Drs. Nigel Fraser and Dangsheng Li for their instructive comments on the manuscript. Y.B.C., C.P.Y. and D.D.W. wrote the manuscript.

## Competing interests

The authors declare no competing interests.
