## [Peer Review File · Nature Communications]

Reviewers' comments:

Reviewer #1 (Remarks to the Author):

This is a quite interesting manuscript. The authors discovered that YTHDF1 is one gene that showed lower expression in Tibetan population compared to controls living at lower altitude. Further studies suggest that YTHDF1 may play roles in hypoxia response. The authors went on to show YTHDF1 promotes NSCLC proliferation and xenograft tumor formation. There are a ton of really interesting data in this report. I think the work deserves consideration for publication in Nat. Communication. I have a few comments that would need to be addressed:

1. There is a lack of any studies to connect effects of YTHDF1 with m6A on mRNA. The authors need to perform a few m6A-seq as well as translation analysis to connect the observed phenotypes with m6A methylation.
2. The cancer part can be confusing as the authors presented two "opposing" effects of YTHDF1. The authors can clarify this as the first tumor promotion effect seems to be intrinsic to YTHDF1. The authors showed effects but lack explanation. Which pathways and what groups of mRNA m6A are recognized by YTHDF1 and how that mediates tumor growth?
3. The drug response part appears more to be a stress response pathway. Again, connect with m6A is needed.
4. Citation seems to be random. m6A cancer papers were not adequately cited or discussed. New insights on m6A in cancer and functions of YTHDF1 were not discussed and connected with the current work.

Reviewer #2 (Remarks to the Author):

Major Issues

The association of YTHDF1 downregulation with cisplatin resistance is intriguing, however it is unclear why the YTHDF1 knockdown tumors would grow faster when treated with cisplatin than when given PBS alone. Figure 4g clearly shows a marked increase in tumor growth in these tumors with cisplatin treatment. The authors claim that cisplatin is increasing tumor cell proliferation under these conditions, but Figure 4i does not show proliferation for these tumors when treated with PBS. This counterintuitive finding and the mechanism driving it should be elucidated by the authors. The association with cisplatin and ROS generation suggests they should try a second therapy that either is or is not driven by ROS, such as radiation or a chemotherapeutic utilizing an alternate mode of action.

Recently published work on hepatocellular carcinoma has associated overexpression of YTHDF1 with poor prognosis, the opposite of the finding here for NSCLC. This result should be mentioned and discussed.

Minor Issues

Line 77: "reducing HIF-dependency reducing roles" ???

Line 87: The introduction is scattered and difficult to follow the overarching line of reasoning. The second paragraph shifts from discussion of hypoxia to epigenetics, providing background with no description of the relevance to hypoxia other than a casual mention of studies examining epigenetics associated with hypoxia. The third paragraph focuses on ROS and anti-oxidant

systems, again with little mention of why these topics are being addressed. In general, the introduction gives excessive detail and should be more focused on the study at hand.

Figure 1: The terminology used in the figure is poorly defined, shifts, and is difficult to follow. For example, "census" refers to Cancer Gene Census, which is referred to in the text as CGC. "Plain" in panel e refers to lowland cattle, while "Tibetan" refers to highland cattle. PSG is undefined. The terminology should be made as consistent as possible to improve readability.

Figure 1b is poorly described in the caption, it is unclear what the multiple rows for each species refer to.

Figures 1c and 1d are also unexplained.

Figure 1e needs controls showing other genes that are consistently transcribed between comparable highland and lowland species.

Figure 2a needs to have the labels for each lane defined.

Line 312: It is unclear where the data showing the decrease in tumors larger than 1 mm is shown. Moreover, the authors should do the experiment with and without deletion of p53 in order to demonstrate conclusively whether YTHDF1 loss inhibits tumor growth in both contexts.

Figure 4b shows a large difference in the plotted bars, but this appears to be driven largely by the fact that more samples were available from cancer than from normal tissue. This should be replotted to better depict the fraction of positive/negative staining in normal and cancerous tissues.

Line 339: "found that 441 (or 21) out of 487 NSCLC patients" ???

Figure 6: The caption for this figure is confusing. The figure does not appear to depict up or downregulation of YTHDF1, as described in the caption, but only differential effects under normoxia or hypoxia.

Response to referees

We thank the reviewers greatly for their thorough reviews and highly appreciate their comments and suggestions, which significantly improved the quality of the manuscript. Please find below a detailed point to point response to each comment. The revised information shown in Page and reciprocal Figures was marked in red below.

Reviewer #1 (Remarks to the Author):

This is a quite interesting manuscript. The authors discovered that YTHDF1 is one gene that showed lower expression in Tibetan population compared to controls living at lower altitude. Further studies suggest that YTHDF1 may play roles in hypoxia response. The authors went on to show YTHDF1 promotes NSCLC proliferation and xenograft tumor formation. There are a ton of really interesting data in this report. I think the work deserves consideration for publication in Nat. Communication. I have a few comments that would need to be addressed:

Response: We thank this reviewer for the positive comment that our work deserves consideration for publication in Nat Communications.

1. There is a lack of any studies to connect effects of YTHDF1 with m6A on mRNA. The authors need to perform a few m6A-seq as well as translation analysis to connect the observed phenotypes with m6A methylation.

Response: We thank this reviewer for raising this important point. As suggested, we did the following studies including m⁶A-seq:

- 1) We did m⁶A-seq in A549 cells.
- 2) We sequenced RNA obtained from the immuno-purified complex of YTHDF1 (RIP-seq) to reveal YTHDF1 bound mRNAs, where 3,676 genes were shared (m⁶A-seq+RIP-seq) as high-confident targets of YTHDF1 (Fig. 2n, Supplementary Table 5), and were mapped to cell cycle and tumor (including lung cancer) related signaling pathways in the KEGG (Kyoto Encyclopedia of Genes and Genomes) pathway database (Fig. 2o, Supplementary Table 5).
- 3) We showed that m⁶A peaks as well as YTHDF1 binding enrichment were observed in CDK2 and CDK4 as shown by Integrative Genomics Viewer (IGV) software (Fig. 2p).
- 4) Furthermore, we showed that knockdown of YTHDF1 in A549 cells significantly reduced CDK2 and cyclin D1 mRNAs in the translating pool (Extended Data Fig. 2i). Page 8-9.

2. The cancer part can be confusing as the authors presented two "opposing" effects of YTHDF1. The authors can clarify this as the first tumor promotion effect seems to be intrinsic to YTHDF1. The authors showed effects but lack explanation. Which pathways and what groups of mRNA m6A are recognized by YTHDF1 and how that mediates tumor growth?

Response: We thank this reviewer for pointing this out. In the revised manuscript, we showed that the mRNAs for many cell cycle checkpoint regulators, including CDK2 and 4 are modified by m⁶A (Fig. 2n-p, Supplementary Table 5). The translational efficiencies of these genes were also downregulated upon YTHDF1 knockdown based on their polysomal profiling (Extended Data Fig. 2i). CDK2, CDK4 and cyclin D1 proteins were reduced in YTHDF1 knockdown cells, with no observed differential mRNA expression (Fig. 2i, m, Extended Data Fig. 2d, e, f). Therefore, under normoxia condition, YTHDF1 is highly expressed in tumor cells to promote cell proliferation through its bound m⁶A modified CDKs. **Page 8-9.**

In the revised version, we found that m⁶A but not the gene expression level in A549/DDP was different from that in A549 cells, and the m⁶A modification level of YTHDF1 bound Keap1 transcript was reduced (Extended Data Fig. 5h, 5i and Supplementary Table 7). Consistently, we showed that knockdown of YTHDF1 in A549 cells reduced Keap1 mRNAs in the translating pool upon DDP treatment (Extended Data Fig. 5j). Therefore, mechanistic studies indicate that ablation of YTHDF1 decreases translational efficiency of the m⁶A-modified Keap1 transcript in an oxidative stress state induced by DDP, which in turn activates the antioxidant ROS clearance system (Nrf2-AKR1C1) leading to DDP resistance and a worse clinical outcome for NSCLC patients after platinum based chemotherapy (Fig. 5 and Extended Data Fig. 5). **Page 11-14.**

3. The drug response part appears more to be a stress response pathway. Again, connect with m6A is needed.

Response: We thank the reviewer for pointing this out. In the revised version, we found that the m⁶A but not the gene expression level in A549/DDP was significantly different from that in A549 cells, and the m⁶A modification level of YTHDF1 bound Keap1 transcript was significantly reduced (Extended Data Fig. 5h, 5i and Supplementary Table 7). Consistently, we showed that knockdown of YTHDF1 in A549 cells reduced Keap1 mRNAs in the translating pool upon DDP treatment (Extended Data Fig. 5j). **Page 13.**

4. Citation seems to be random. m6A cancer papers were not adequately cited or discussed. New insights on m6A in cancer and functions of YTHDF1 were not discussed and connected with the current work.

Response: We appreciate this comment and have updated the discussion and references on m⁶A in cancer and functions of YTHDF1. **Page 14-17.**

Reviewer #2 (Remarks to the Author):

Major Issues

The association of YTHDF1 downregulation with cisplatin resistance is intriguing, however it is

unclear why the YTHDF1 knockdown tumors would grow faster when treated with cisplatin than when given PBS alone. Figure 4g clearly shows a marked increase in tumor growth in these tumors with cisplatin treatment. The authors claim that cisplatin is increasing tumor cell proliferation under these conditions, but Figure 4i does not show proliferation for these tumors when treated with PBS. This counterintuitive finding and the mechanism driving it should be elucidated by the authors.

Response: We apologize for the confusion. In our original Figure 4g~4i, YTHDF1 knockdown with scramble shRNA control cells were injected in nude mice with different cell numbers in order to make sure that the tumors reached about 50mm³ in size simultaneously before treatment with PBS or DDP. Therefore, more YTHDF1 knockdown cells were injected compared with scramble shRNA control cells, due to reduced cell proliferation rate both in vivo and in vitro. This has contributed to the above confusion mentioned above.

To validate this phenotype again, we repeated the xenograft tumor formation assay, and found that for A549 xenografts, control tumors treated with PBS grew to average 470, 253 and 145 mm³ in control shRNA, YTHDF1 shRNA #1 and shRNA#2 groups, respectively, 21 days following randomization (Fig. 4g). Interestingly, scramble shRNA tumors treated with DDP grew to ~23% of PBS treated tumor size, however, YTHDF1 knockdown tumors did not show significant difference in tumor size when we compared PBS and DDP treatment groups. (Fig. 4g-i, Extended Data Fig. 4b-e). These revised data is consistent with H1299 xenografts shown in original Extended Data Fig. 4c-e.

In addition, the original text “In the DDP treatment tumor groups, dramatically higher proliferation as measured by Ki67 IHC...” means that YTHDF1 knockdown tumor cells proliferate faster than scramble shRNA cells in DDP treatment group, we were not comparing the proliferation rate of YTHDF1 knockdown tumors in PBS group with that in DDP group. Therefore, we did not intend to conclude that DDP promotes tumor cell proliferation. Furthermore, the misleading text in the revised version was also corrected. **Page 11.**

The association with cisplatin and ROS generation suggests they should try a second therapy that either is or is not driven by ROS, such as radiation or a chemotherapeutic utilizing an alternate mode of action.

Response: We understand this reviewer’s concern. To corroborate this phenotype, we also treated the NSCLC cancerous cells with radiation or navitoclax, an inhibitor of the anti-apoptotic factors BCL-xL and BCL-2¹. We found that YTHDF1 knockdown inhibited cellular apoptosis in radiation but not navitoclax treatment group (Extended Data Fig. 5g). **Page 11.**

Recently published work on hepatocellular carcinoma has associated overexpression of YTHDF1 with poor prognosis, the opposite of the finding here for NSCLC. This result should be mentioned and discussed.

Response: We thank this reviewer's for pointing this out. The study mentioned by this reviewer showed that YTHDF1 was upregulated in hepatocellular carcinoma (HCC), which correlated with significant shorter OS and DFS survival rate ². This is different from our observation that YTHDF1 low expression correlates with worse NSCLC clinical outcome by IHC analysis using tissue microarray. This apparent discrepancy suggests that we need to take both the protein and mRNA expressions of YTHDF1 into consideration to characterize its functional roles in future. In addition, the m⁶A modified mRNA targets of YTHDF1 could be time- and cellular context- dependent, which leads to the differential functions of YTHDF1 in various cancers progression. As suggested by this reviewer, we have discussed and cited this in the revised version. **Page 17.**

Minor Issues

Line 77: "reducing HIF - dependency reducing roles" ???

Response: We have modified the original sentences as following "Therefore, hypoxia adaptation selected genes more likely play anti-hypoxia or anti-HIF1/2 dependent roles to make animals or humans behave normally under hypoxic environments". **Page 3.**

Line 87: The introduction is scattered and difficult to follow the overarching line of reasoning. The second paragraph shifts from discussion of hypoxia to epigenetics, providing background with no description of the relevance to hypoxia other than a casual mention of studies examining epigenetics associated with hypoxia. The third paragraph focuses on ROS and anti-oxidant systems, again with little mention of why these topics are being addressed. In general, the introduction gives excessive detail and should be more focused on the study at hand.

Response: We thank the reviewer for this valuable suggestion. In the revised version, we have condensed the introduction and focused on the study at hand. **Page 3-4, 11-12.**

Figure 1: The terminology used in the figure is poorly defined, shifts, and is difficult to follow. For example, "census" refers to Cancer Gene Census, which is referred to in the text as CGC. "Plain" in panel e refers to lowland cattle, while "Tibetan" refers to highland cattle. PSG is undefined. The terminology should be made as consistent as possible to improve readability.

Response: We have corrected the misleading labels both in the text and figures. The "census" is replace by "CGC", "plain" is replaced "lowland cattle", "Tibetan" is replaced by "highland cattle", PSG is defined as "positive selected genes" in the figure legend. **Fig. 1a, 1b, 1e, Extended Data Fig. 1a, 1b. Page 5, 29-30.**

Figure 1b is poorly described in the caption, it is unclear what the multiple rows for each species refer to.

Response: Per this reviewer's suggestion, detail descriptions for the multiple rows in Fig. 1b were added in the revised version. **Page 29-30.**

Figures 1c and 1d are also unexplained.

Response: Details descriptions for Figure 1c and 1d were added in the revised version. **Page 29-30.**

Figure 1e needs controls showing other genes that are consistently transcribed between comparable highland and lowland species.

Response: We thank this reviewer for this valuable suggestion. In the revised manuscript, we show that the other two YTH domain family members YTHDF2 and YTHDF3 are transcribed at similar levels. We have also normalized the relative mRNA expression for YTHDF1, YTHDF2 and YTHDF3 to cattle GAPDH mRNA expression (**Fig. 1e, Supplementary Table 8**). **Page 7.**

Figure 2a needs to have the labels for each lane defined.

Response: Per this reviewer's suggestion, detail descriptions for Figure 2a were added in the revised version (**Fig. 2a**). **Page 31-32.**

Line 312: It is unclear where the data showing the decrease in tumors larger than 1 mm is shown. Moreover, the authors should do the experiment with and without deletion of p53 in order to demonstrate conclusively whether YTHDF1 loss inhibits tumor growth in both contexts.

Response: We understand the reviewer's concern. As suggested by this reviewer, the original text was replaced by "As compared with reciprocal control KP or K mutant mice, KPY or KY mice showed a dramatic decrease in tumor burden as measured by tumor number (large tumors, $\geq 1\text{mm}^2$) and tumor size, respectively, indicating that YTHDF1 promotes the lung tumor progression driven by KRAS with or without p53 mutation (**Fig. 3g, h, Extended Data Fig. 3e, f**)." **Page 9-10.**

NSCLC cancerous cell line H1299 has been demonstrated to be a p53-deficiency cell line (NCI-H1299 (ATCC® CRL-5803™), in line with another cell line A549 expressing wild-type p53, we provided evidences showing that YTHDF1 knockdown in both cell lines reduced cell proliferation, colony formation and xenograft tumor formation abilities (**Fig. 2, 4 and Extended Data Fig. 2, 4**).

In addition, we demonstrated that YTHDF1 depletion restrains *de novo* lung adenocarcinomas (ADC) progression in *Kras*^{G12D} with or without *Trp53*^{lox/lox} (KP or K) lung cancer mouse models in vivo (**Fig. 3 and Extended Data Fig. 3**). Therefore, we concluded that loss of YTHDF1 inhibits tumor growth in both contexts. **Page 9-10.**

Figure 4b shows a large difference in the plotted bars, but this appears to be driven largely by the fact that more samples were available from cancer than from normal tissue. This should be replotted to better depict the fraction of positive/negative staining in normal and cancerous tissues.

Response: We thank this reviewer for this valuable suggestion. In the revised version, we added 50 more non-cancerous lung tissues (NCLT), and found that the positive YTHDF1 expression percentage is still about 42%, which is not dramatically affected by the number of tissue samples. Therefore, we replotted the data with more NCLT samples (**Fig. 4b**). **Page 10.**

Line 339: “found that 441 (or 21) out of 487 NSCLC patients” ???

Response: We have corrected the misleading writing in the revised version as “found that 462 NSCLC patients were treated by platinum based chemo-(441/462) or radio-(21/462) therapy alone, whereas 25 patients were treated by both chemo- and radio-therapies (**Supplementary Table 6**).” **Page 10.**

Figure 6: The caption for this figure is confusing. The figure does not appear to depict up or downregulation of YTHDF1, as described in the caption, but only differential effects under normoxia or hypoxia.

Response: We thank this reviewer for this valuable suggestion, and we have added the up- or downregulation of YTHDF1 under different conditions in the revised **Figure 6**. **Page 36-37.**

Reviewer #3 comments to authors

This study describes an analysis of cancer-related genes in humans and other domestic animals that are adapted to the hypoxic conditions found at the Tibetan high-altitude plateau. The authors integrated different methods to detect positive selection at the population level (using the method described in Grossman et al 2013) and applied it to cancer-related genes from two databases. This analysis revealed an overall rapid evolution of these genes and cases of positive selection. The two highest-ranked genes are YTHDF1 and TEX2. The role of YTHDF1 expression level in tumor development and chemotherapy outcome was subsequently characterized in great detail, providing new insights into cancer and potentially therapy. Overall, this is a very interesting manuscript that uses population genetic data of multiple naturally-adapted species to find novel genes implicated in cancer. Hence, I recommend publishing it after a few points are addressed.

Response: We thank this reviewer for the positive comment that our work deserves consideration for publication in Nat Communications.

Major comments

* "We found that differentially expressed genes were significantly enriched in categories associated with cancer, cell death and apoptotic processes (Supplementary Table 1).

While cell death and apoptosis are GO terms that appear at the top after sorting by P-value, there are no direct cancer-related GO terms. These enrichments should be better described. The table should also be sorted by P-value.

Response: We thank this reviewer for raising this important point. We have revised this description as: We found that differentially expressed genes were significantly enriched in categories associated with cell death, apoptosis, migration, etc., many of these genes have been documented to play important roles during tumorigenesis (**Supplementary Table 1**). As suggested, we revised Supplementary Table 1, and the table was also sorted by P-value. **Page 6.**

* "no sense mutations within YTHDF1 were identified". Does that refer to the human data or also the pairs of lowland/highland domestic mammals? If sense mutation refers to synonymous mutations, I wonder if there are any non-synonymous mutations?

Response: We apologize for the confusion. Our original description "no sense mutation" refers to "no amino acid change or no non-synonymous mutation", and the sequence analysis was performed in highland cattle. Furthermore, the misleading text in the revised version was also corrected as "no amino acid change within YTHDF1 was identified in highland cattle." **Page 6-7.**

Minor comments:

* This link <http://hgdp.uchicago.edu/software/> does not work.

Response: We have corrected the misleading writing in the revised version as <http://hgdp.uchicago.edu/Software/>. **Page 18.**

Page 3: "reducing HIF- - dependency reducing roles" This is confusing.

Response: We have modified the original sentences as following "Therefore, hypoxia adaptation selected genes more likely play anti-hypoxia or anti-HIF1/2 dependent roles to make animals or humans behave normally under hypoxic environments". **Page 3.**

Page 5: "a positively selected candidate genes"  "gene"

Response: We have modified the original writing as following "a positively selected candidate gene". **Page 4.**

Typo in Figure 1B 'P-value'

Response: We have corrected the typo in Figure 1b “P-value”. Fig. 1b.

References

1. Tan, N. *et al.* Navitoclax enhances the efficacy of taxanes in non-small cell lung cancer models. *Clinical cancer research : an official journal of the American Association for Cancer Research* **17**, 1394-1404 (2011).
2. Zhao, X. *et al.* Overexpression of YTHDF1 is associated with poor prognosis in patients with hepatocellular carcinoma. *Cancer biomarkers : section A of Disease markers* **21**, 859-868 (2018).

REVIEWERS' COMMENTS:

Reviewer #1 (Remarks to the Author):

The authors have made efforts to address my comments. The work is more coherent. There are many other questions but I think those can be studied in future research.

Reviewer #2 (Remarks to the Author):

The authors have responded well to all previous critiques. I feel the manuscript is now suitable for publication in Nature Communications.

Reviewer #3 (Remarks to the Author):

The authors have addressed all my concerns. Hence I recommend accepting this manuscript.

Reviewer #4 (Remarks to the Author):

Shi et al. proposed a study to investigate the underlying genetic component of hypoxia and its consequences under normal and pathological conditions. By comparing Tibetans and Tibetan mammals with their low elevation counterparts, the authors identified a candidate gene, YTHDF1, to be a regulator for both hypoxia adaptation and cancer progression. The functional role of this gene was then investigated and discussed in great detail. Importantly, the authors showed that knockout of YTHDF1 gene is associated with inhibited cancer cell proliferation and colony formation. Additionally, they proposed that YTHDF1 gene could mediate a cell's response to DDP-dependent chemotherapy in non-small cell lung cancer which could provide insights into the development of improved therapeutic strategies.

In sum, the manuscript is well written, and the logic is easy to follow. The topic and the idea of using hypoxia adaptation to approach oncogenesis genes are expected to be of interest to a wide range of audience. The manuscript reported many interesting findings and are well-supported by in vitro, in-vivo, and computational methods and results. In addition, the authors have well-addressed the comments from previous reviewers.

I just have some minor suggestions which are listed below:

Page 9, line 298: 'large tumors ...' should be following 'tumor size' instead of 'tumor number'.

Page 19, line 603: 'tag' should be TAG, which was used in other sections of the manuscript.

Page 19, line 604-608: to increase reproducibility, the versions of different genomes should be provided.

Response to referees

We thank the reviewers greatly for their thorough reviews and highly appreciate their comments and suggestions, which significantly improved the quality of the manuscript. Please find below a detailed point to point response to each comment. The revised information shown in Page and reciprocal Figures was marked in red below.

Reviewer #1 (Remarks to the Author):

The authors have made efforts to address my comments. The work is more coherent. There are many other questions but I think those can be studied in future research.

Response: We thank this reviewer for the positive comment.

Reviewer #2 (Remarks to the Author):

The authors have responded well to all previous critiques. I feel the manuscript is now suitable for publication in Nature Communications.

Response: We thank this reviewer for the positive comment.

Reviewer #3 (Remarks to the Author):

The authors have addressed all my concerns. Hence I recommend accepting this manuscript.

Response: We thank this reviewer for the positive comment.

Reviewer #4 (Remarks to the Author):

Shi et al. proposed a study to investigate the underlying genetic component of hypoxia and its consequences under normal and pathological conditions. By comparing Tibetans and Tibetan mammals with their low elevation counterparts, the authors identified a candidate gene, YTHDF1, to be a regulator for both hypoxia adaptation and cancer progression. The functional role of this gene was then investigated and discussed in great detail. Importantly, the authors showed that knockout of YTHDF1 gene is associated with inhibited cancer cell proliferation and colony formation. Additionally, they proposed that YTHDF1 gene could mediate a cell's response to DDP-dependent chemotherapy in non-small cell lung cancer which could provide insights into the development of improved therapeutic strategies.

In sum, the manuscript is well written, and the logic is easy to follow. The topic and the idea of using hypoxia adaptation to approach oncogenesis genes are expected to be of interest to a wide

range of audience. The manuscript reported many interesting findings and are well-supported by in vitro, in-vivo, and computational methods and results. In addition, the authors have well-addressed the comments from previous reviewers.

Response: We thank this reviewer for the positive comment that we have well-addressed the comments from previous 3 reviewers.

I just have some minor suggestions which are listed below:

Page 9, line 298: 'large tumors ...' should be following 'tumor size' instead of 'tumor number'.

Response: We thank this reviewer for pointing this out, and have moved “large tumors...” after tumor size. Page 9.

Page 19, line 603: 'tag' should be TAG, which was used in other sections of the manuscript.

Response: We thank this reviewer for pointing this out, and have changed “tag” to “TAG” in the revised version. Page 17.

Page 19, line 604-608: to increase reproducibility, the versions of different genomes should be provided.

Response: We appreciate this comment and have added the version “...were downloaded from Ensembl (version 72) by BioMart...”. Page 17-18.